# A closed-loop auditory stimulation approach selectively modulates alpha oscillations and sleep onset dynamics in humans

Henry Hebron[1,2,3]*, Beatrice Lugli[1], Radost Dimitrova[1], Valeria Jaramillo[1,2,3], Lisa R. Yeh[1], Edward Rhodes[4,5], Nir Grossman[4,5‡], Derk-Jan Dijk[2,3‡], Ines R. Violante[1‡]*

**1** School of Psychology, University of Surrey, Guildford, United Kingdom, **2** Surrey Sleep Research Centre, University of Surrey, Guildford, United Kingdom, **3** UK Dementia Research Institute Care Research and Technology Centre, Imperial College London and the University of Surrey, Guildford, United Kingdom, **4** Department of Brain Sciences, Imperial College London, London, United Kingdom, **5** UK Dementia Research Institute Imperial College London, United Kingdom

‡ These authors are joint senior authors on this work.
* h.hebron@surrey.ac.uk (HH); ines.violante@surrey.ac.uk (IRV)

**Data Availability Statement:** All data required for (re)production of figures are available from the first author's github repository ( https://github.com/

## Abstract

Alpha oscillations play a vital role in managing the brain's resources, inhibiting neural activity as a function of their phase and amplitude, and are changed in many brain disorders. Developing minimally invasive tools to modulate alpha activity and identifying the parameters that determine its response to exogenous modulators is essential for the implementation of focussed interventions. We introduce Alpha Closed-Loop Auditory Stimulation (αCLAS) as an EEG-based method to modulate and investigate these brain rhythms in humans with specificity and selectivity, using targeted auditory stimulation. Across a series of independent experiments, we demonstrate that αCLAS alters alpha power, frequency, and connectivity in a phase, amplitude, and topography-dependent manner. Using single-pulse-αCLAS, we show that the effects of auditory stimuli on alpha oscillations can be explained within the theoretical framework of oscillator theory and a phase-reset mechanism. Finally, we demonstrate the functional relevance of our approach by showing that αCLAS can interfere with sleep onset dynamics in a phase-dependent manner.

## Introduction

The alpha rhythm (approximately 10 Hz) is a defining electrophysiological feature of the waking human brain, most prominent in parietal and occipital areas [1], but observed in various neural regions [2,3]. Alpha oscillations have been associated with fundamental processes, from memory [4] and perception [5–8], to ageing [9,10] and disease [11–13]. However, selective modulation of the alpha rhythm remains challenging, limiting our knowledge of the underlying determinants of the alpha oscillation's response to stimuli and the possible functional consequences of its selective modulation.

One promising route to efficacious alpha modulation is to address this rhythm on its own terms, using a closed-loop approach which leverages the instantaneous features of the

HHebron/aclas_PB) and on zenodo (http://doi.org/10.5281/zenodo.1084843).

**Funding:** H.H. is supported by the University of Surrey's Doctoral College Scholarship Award. VJ is supported by the Swiss National Science Foundation (P2EZP3_199918, P500PB_217827). N.G. is supported by the UK Dementia Research Institute through UK DRI Ltd, principally funded by the UK Medical Research Council, and additional funding partner Alzheimer's Society., Wellcome Trust fellowship (097443/Z/11/Z), Science & PINS Award for Neuromodulation, and NIHR IBRC Confident in Concept Award. D.-J.D is supported by the UK Dementia Research Institute [award number UKDRI-7005.] through UK DRI Ltd, principally funded by the UK Medical Research Council, and additional funding partner Alzheimer's Society. I.R.V. is supported by the Biotechnology and Biological Sciences Research Council (BB/S008314/1). The funders played no role in the study design, data collection and analysis, decision to publish, or preparation of the manuscript.

**Competing interests:** N.G. is part in a patent application on the ecHT technology, assigned to MIT, and is a founder in a company that utilises it.

**Abbreviations:** αCLAS, Alpha Closed-Loop Auditory Stimulation; AASM, American Academy of Sleep Medicine; AEP, auditory-evoked potential; ASR, automatic subspace reconstruction; ECG, electrocardiogram; ecHT, endpoint corrected Hilbert transform; EMG, electromyography; EOG, electro-oculogram; IAF, individual alpha frequency; iEEG, intracranial electroencephalographic; ISI, inter-stimulus interval; KDT, Karolinska drowsiness test; KSS, Karolinska sleepiness scale; LMEM, linear mixed-effects model; MEP, motor-evoked potential; NREM, non-rapid eye movement; PLI, phase lag index; PLV, phase-locking value; PSQI, Pittsburgh Sleep Quality Index; PRC, phase response curve; PTC, phase transfer curve; ROI, region of interest; TMS, transcranial magnetic stimulation; TST, total sleep time.

oscillation in real-time. Influential theories posit that neuronal excitability fluctuates as a function of alpha phase and amplitude [14,15], since neurons more readily fire during the trough of alpha oscillations, and during periods of lower alpha amplitude [2,3,16,17]. Accordingly, the amenability of alpha to exogenous influence should itself be phase-dependent, owing to the interdependent nature of spikes and oscillations [18].

In this vein, alpha phase dependencies have been investigated in visual perception [5,8]. These studies typically sort the EEG, post hoc, on the (binary) basis of perception of a visual stimulus, finding an effect of alpha phase at stimulus onset. However, many time-frequency approaches provide phase estimates dependent on what proceeds them (i.e., they are non-causal), which might drive spurious correlations [19]. Furthermore, conscious perception itself may have an electrophysiological component which backwards-influences the preceding phase estimate [20,21]. Experiments in which specific phases of the alpha oscillation are targeted in real-time are therefore needed to clarify phase-response dependencies.

Phase dependency of the response to stimuli has been demonstrated in a real-time manner for the transcranial magnetic stimulation (TMS) motor-evoked potential (MEP) [22–25]. This approach circumvents the issue of time-dependency, but TMS pulses cause large artefacts in the EEG, rendering rapidly repeating closed-loop (see [26]) stimulation unfeasible, and are clouded by concomitant residual auditory and somatosensory activity resulting from the TMS pulse [27]. Therefore, previous studies have not been able to provide neurophysiological characterisations of phase dependencies of the alpha rhythm, i.e., whether alpha oscillations themselves respond to stimuli in a phase-dependent manner. Closed-loop sensory stimulation has the potential to overcome this limitation, since sensory stimuli do not create artefacts in the EEG (allowing repeated neurophysiological interrogation), with the additional advantage of utilising safe, easy to apply, and affordable technology. This approach has been used in sleep studies in which sounds are commonly phase-locked to the slow oscillations (0.5 to 4 Hz) of sleep [28–34]. Some of these studies have found a phase-dependent effect of stimuli whereby sounds boost or diminish slow oscillations on a phase-dependent basis [28–34], although this remains contentious [30]. However, the closed-loop approaches used for slow oscillations cannot be directly employed to target the much faster alpha oscillation, since those approaches predominantly operate on the basis of an amplitude-threshold, which is unsuitable for alpha rhythms because these are highly variable in their amplitude and are instead defined on the basis of frequency. Recently, strategies have been proposed to overcome this limitation, either relying on forecasting algorithms for real-time phase prediction [22] or through the application of a causal band-pass filter, the endpoint corrected Hilbert transform (ecHT) [35]. The ecHT is a method that tracks the instantaneous phase and envelope amplitude of an oscillatory signal, allowing the instantaneous phase to be reliably computed in real-time [19,35].

Here, we introduce Alpha Closed-Loop Auditory Stimulation (αCLAS), which uses the ecHT approach [35] to administer sounds phased-locked to alpha rhythms. We first explored αCLAS ability to induce phase-dependent modulations to the alpha rhythm in 2 cortical locations with comparatively different endogenous alpha power during rest, eyes-closed conditions (i.e., frontal–lower and parietal–higher alpha power, [1]; experiments 1 and 2, respectively). We observed that repetitive-αCLAS induced phase-dependent modulations of alpha power, frequency, and connectivity in a spatially specific manner. To further understand these effects, we used single-pulse αCLAS (experiments 3 and 4). This approach confirmed that the effects of αCLAS are indeed dependent on the endogenous amplitude of alpha oscillations and can be explained using oscillator theory and a phase-reset model. Finally, we applied these principles during the transition to sleep, a process in which alpha oscillations become naturally dampened (experiment 5). We showed that the dynamics of sleep onset can be modulated in a phase-dependent manner, such that αCLAS administered prior to the troughs of alpha

oscillations resulted in "shallower" sleep. This was characterised by a reduction in the amount of non-rapid eye movement (NREM) sleep stages 2/3 (i.e., N2+) and longer latencies for N2 + sleep, accompanied by a decrease in alpha frequency and a limited reduction in alpha activity.

## Results

### Accuracy and spatial specificity of alpha closed-loop auditory stimulation

We first tested whether αCLAS could successfully deliver phased-locked stimuli to the resting alpha oscillation at different cortical locations. It is well established that during rest with eyes-closed alpha power is typically higher in parietal and occipital areas than frontal regions [1]. We administered sounds repeatedly at 4 orthogonal phases: pre-peak 330˚, post-peak 60˚, pre-trough 150˚, post-peak 240˚, **Fig 1A**. Phase was computed in real-time (using the endpoint-corrected Hilbert Transform, ecHT [35]) based on the EEG activity recorded from one electrode in the αCLAS EEG system placed over the frontal, Fz (experiment 1) or parietal cortex, Pz (experiment 2). Simultaneously, we recorded whole-scalp activity using high-density (hd) EEG from a separate EEG system (128 channels). Participants were exposed to 30-s of repeated sounds (termed "on"), interleaved with 10-s of silence ("off"), while seated with their eyes closed (experiment 1, $N = 23$, experiment 2, $N = 28$; 10 blocks per phase per experiment, **Fig 1B**).

The presence of a spectral peak in the alpha band (7.5 to 12.5 Hz) during the "off" period was first confirmed in the phase-locking electrode from the αCLAS EEG system. A peak was observed in both the group average and majority of participants (experiment 1: 21/23; experiment 2: 28/28 participants, **Fig 1C**), demonstrating that alpha activity was present at the target electrode. The individual alpha frequency (IAF) in the "off" periods correlated strongly with the average inter-stimulus interval (ISI) of the αCLAS sounds, indicating that stimulation frequency was tailored to the individual, with approximately 1 stimulus applied per alpha cycle (**Fig 1D**). We then examined the phase-locking accuracy of the αCLAS EEG system by calculating the mean resultant vector length and average angle of phase at stimulus onset (for every stimulus delivered during αCLAS "on" periods), resulting in a value between 0 (uniform distribution) and 1 (unimodal distribution, perfect phase-locking). The resultants were high for both experiments and for all targeted phases, with significant unimodal distributions (experiment 1, resultant mean ± std: 0.85 ± 0.05; experiment 2: 0.87 ± 0.04; Z-tests confirm significant unimodality in all conditions; Rayleigh z values >22, ps < 0.001, **Fig 1E**). Furthermore, the average phase angles were highly consistent across participants and orthogonal between conditions (experiment 1, phase angle deviation from target, mean ± std: 2.05˚ ± 1.35; experiment 2: 4.45˚ ± 1.26). We then measured the resultants for the equivalent electrodes in the hd-EEG system (Fz and Pz in the hd-EEG were adjacent to those in the αCLAS EEG system). Likewise, resultants were high in the hd-EEG system (experiment 1, mean ± std: 0.47 ± 0.13; experiment 2: 0.63 ± 0.13; z values >20, ps < 0.001, **Fig 1F**) and showed high phase-specificity. The lower resultant vector lengths and slightly different phase angles displayed for the hd-EEG are a consequence of the scalp current density (i.e., Laplacian) reference used here (see **Fig A** in **S1 File** for a comparison between resultants at the target electrode in the hd-EEG system using Laplacian or mastoid reference as in the αCLAS EEG system). Stimulus onset phase angles were comparable across experiments and EEG systems. Finally, we took advantage of the spatial resolution of the hd-EEG to assess the spatial selectivity of the phase-locking. Phase-locking was greatest around the targeted electrodes as indicated by higher resultant values (**Fig 1F** and **Fig B** in **S1 File**).

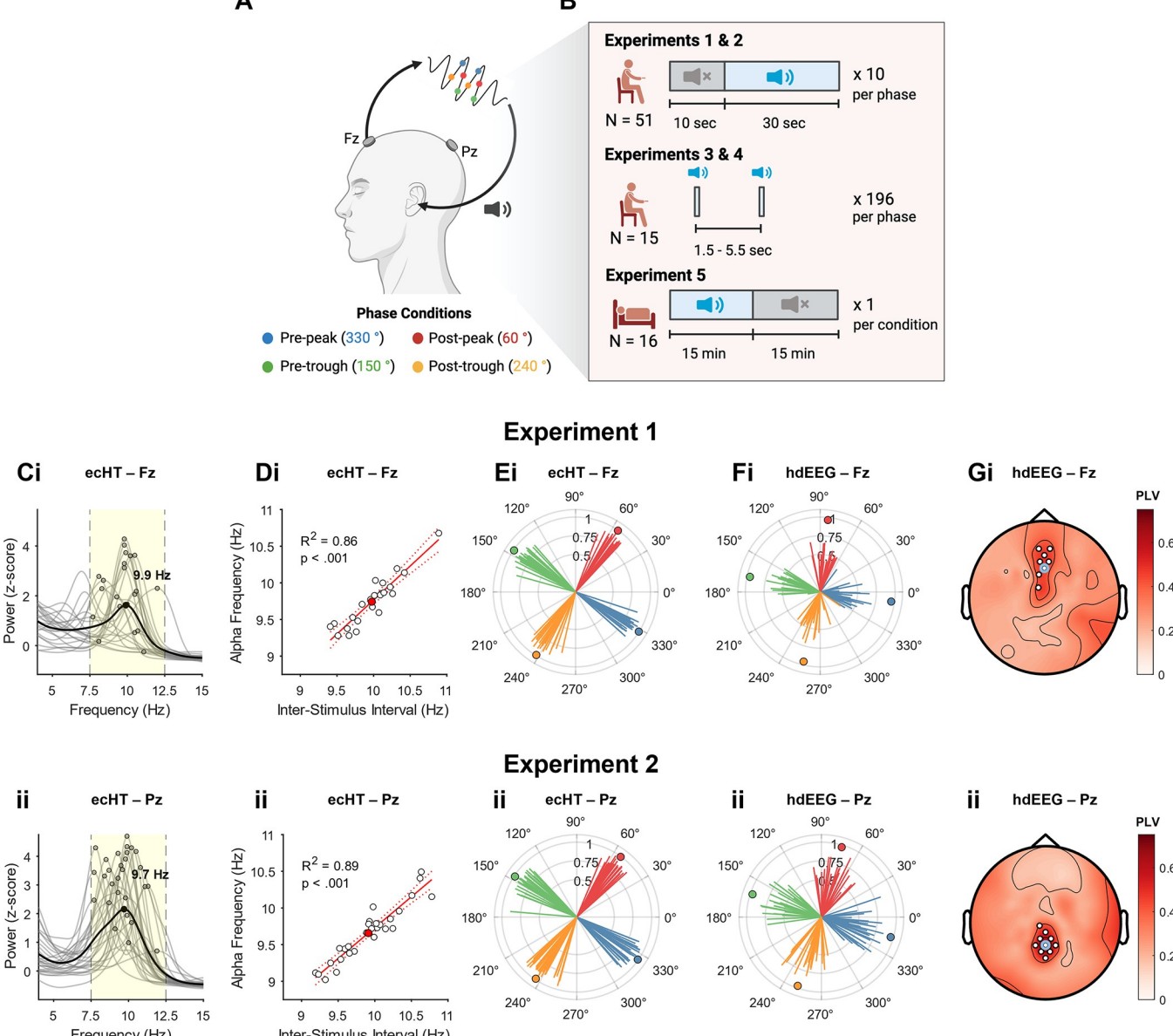

**Fig 1. Experimental design and phase-locking validation. (A)** Schematic of closed-loop auditory stimulation. The signal is processed in real-time to compute its instantaneous phase (coloured dots correspond to the 4 orthogonal phases targeted), and pulses of pink noise (20 ms duration, 80 dB volume) are delivered at the prespecified phase of the target rhythm. Phase information is extracted from the EEG signal at frontal, Fz, or parietal, Pz, locations (Fz and Pz refer to positions on the 10–20 EEG International system). Note: Figure was produced using BioRender. **(B)** Pictorial summary of the 5 experiments conducted, indicating periods of sound "on" stimulation and "off" periods, as well as the overall number of participants and number of trials per phase condition. **(C)** Normalised (z-score; i.e., power each frequency is expressed as SD's from mean power across all frequencies, per participant) power spectrum computed for "off" periods, taken from the phase-locking electrode (from the αCLAS EEG system) in **(i)** experiment 1 (Fz) and **(ii)** experiment 2 (Pz). Phase-locked band (7.5–12.5 Hz) highlighted in yellow. **(D)** Stimulation (Inter-Stimulus Interval; ISI) vs. alpha frequency as computed for "off" periods only, taken from the phase-locking electrode in **(i)** experiment 1 (Fz) and **(ii)** experiment 2 (Pz). White circles represent individual participants, red circle shows group average. Red line shows simple linear model fit to data. **(E)** Average phase angle and resultant at stimulus onset taken from the phase-locking electrode (from the αCLAS EEG system) in **(i)** experiment 1 (Fz) and **(ii)** experiment 2 (Pz). Lines represent individual participants, circles show the group average phase. **(F)** Same as **(E)** but computed using the respective electrodes from the high-density (hd) EEG; **(i)** experiment 1 (Fz) and **(ii)** experiment 2 (Pz). **(G)** Topoplot of resultant—shows the mean PLV (i.e., the mean resultant vector length) for each phase condition at all channels of the hd-EEG data. White marks indicate channels at which PLV >0.4 and *p* < 0.05. Black marks indicate channels at which PLV <0.4 and *p* < 0.05. Light blue circles indicate the approximate position of the target electrode, and *p*-values from FDR-corrected V-test for directionality. Resultants and *p*-values were averaged across 4 phase conditions. Shown for **(i)** experiment 1 and **(ii)** experiment 2. αCLAS, Alpha Closed-Loop Auditory Stimulation; ISI, inter-stimulus interval; PLV, phase-locking value.

## αCLAS induces phase and location-specific effects on alpha power and frequency

Having shown that αCLAS delivers sounds accurately at the prescribed phase in the 2 targeted electrode locations, we next examined whether αCLAS-induced phase-dependent changes to the alpha oscillations we set out to target. We first investigated whether power in the alpha frequency band was differently modulated by the phase at which alpha oscillations were targeted by αCLAS by performing one-way ANOVAs on the hd-EEG data (with log-transformed power ratios of "on" to "off" as the dependent variable and αCLAS phase as the independent variable). We observed a main effect of phase on alpha power when sounds were phase-locked to Fz EEG (experiment 1). This effect was localised to frontal midline regions, around the target electrode (**Fig 2A** shows the topography of the effect of αCLAS phase determined by the ANOVA, for the centre, i.e., 10 Hz, alpha frequency; **Figs C** and **E** in **S1 File** show the specificity of this effect in the alpha band).

To further investigate the effects of phase on alpha power, we performed time-frequency analysis focusing on a cluster of 4 electrodes, which included Fz and the surrounding electrodes (all from the hd-EEG system; see Methods). Changes in power were circumscribed to the alpha band and persisted throughout the 30-s stimulation period (ANOVA showing main effect of phase, **Fig 2Bi**). To better understand this effect, we compared power changes between opposite phases (*t* tests between conditions, **Fig 2Bii-2Biii**). We again observed significant effects of αCLAS phase in the alpha band power that were present throughout the stimulation period, particularly when comparing the post-peak and post-trough conditions (**Fig 2Biii**). Noteworthily, these effects appeared to be frequency-specific, i.e., frequencies within the alpha band were not modulated equally. When different phases were targeted, the power at different frequencies within the alpha band was indeed significantly increased or decreased (**Fig 2C**). The manner in which these power changes were distributed suggested a slowing or quickening of alpha rhythms; in the post-trough condition, for example, power decreased for high alpha frequencies, and increased for low alpha frequencies (i.e., slowing), and the inverse was true for the opposite phase, i.e., post-peak. For pre-peak and pre-trough conditions relative changes in power for specific frequencies appeared more modest, with the former mostly decreasing power in the alpha band and the latter largely increasing it.

To verify whether different phases impacted the frequency of alpha oscillations differently, we estimated instantaneous frequency (within the alpha band) across all hd-EEG channels, following the method proposed by Cohen [36]. The αCLAS-induced frequency change for each condition was then computed by subtracting the average alpha frequency of "off" from "on" periods. The topography of the effect of phase on frequency was assessed with an ANOVA on the hd-EEG data with frequency change as the dependent variable and αCLAS phase as the independent variable (**Fig 2Di**). This showed a strikingly similar distribution to the effect of phase on power (**Fig 2A**). Comparison of the frequency change in the same frontal cluster where we had previously investigated power changes showed significant differences in frequency between phase conditions, in agreement with what was inferred from the power change spectra, i.e., a slowing or quickening of different extents (**Fig 2Dii**, these differences were not present during the "off" period, **Fig G** in **S1 File**). The effects of αCLAS on power and frequency cannot be simply explained by entrainment to a periodic stimulus, as the stimuli delivered by αCLAS was aperiodic (**Fig H** in **S1 File**).

We then performed the same analyses to experiment 2, targeting Pz. Contrary to our previous results, we did not observe a significant effect of αCLAS phase on the power of alpha oscillations (**Fig 2E and 2F** and **Figs D** and **F** in **S1 File**). We observed a significant modulation of alpha power when focusing on the parietal ROI centred around Pz (cluster of 4 electrodes

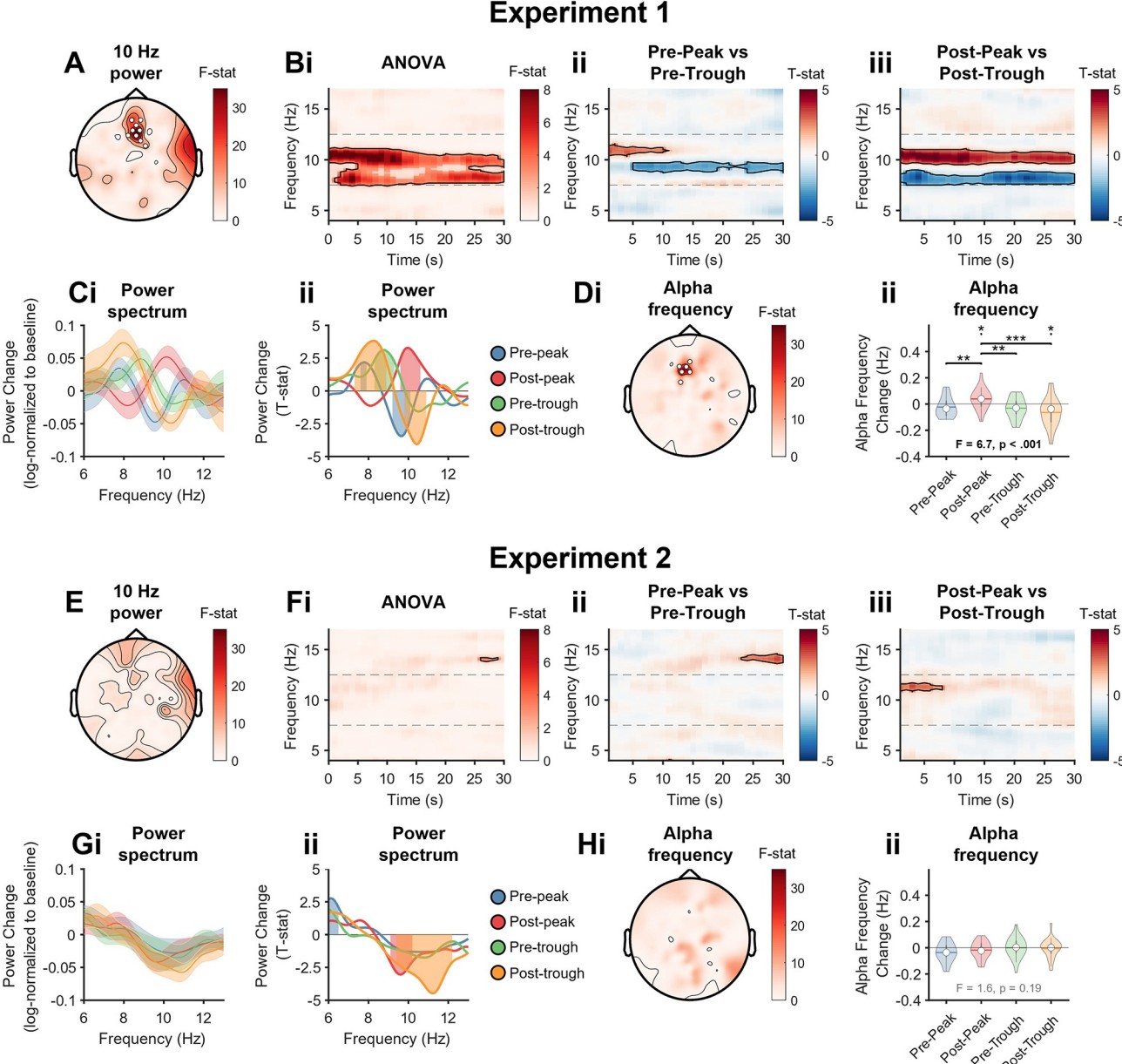

**Fig 2. Stimulation-induced power and frequency changes. (Ai)** Topography of 10 Hz (9.8–10.2 Hz) power change ANOVA. We computed the power spectral density for every phase condition in the "on" and "off" periods and expressed stimulation-induced changes by log-transforming the ratio of "on" to "off" periods per frequency bin. White marks indicate cluster-corrected $p < 0.05$. **(Aii)** Time-frequency ANOVA. Outlined clusters $p < 0.05$. **(Bi)** Time-frequency, pre-peak vs. pre-trough, for the frontal ROI (see ROI specification in Materials and methods). Positive values indicate pre-peak > pre-trough. Outlined clusters ANOVA $p < 0.05$. **(Bii)** Time-frequency, pre-peak vs. pre-trough, at respective regions of interest. Positive values indicate post-peak > post-trough. Outlined clusters $t$ test $p < 0.05$. **(Ci)** Power change spectrum (% change from "off" periods) across the alpha-band frequencies for the 4 targeted phases collapsed across the stimulation period in the frontal ROI. **(Cii)** Power change spectrum (T-values, one-sample $T$ test) at respective region of interest. Shaded regions $p < 0.05$. **(Di)** Topography of frequency change ANOVA. White marks indicate cluster-corrected $p < 0.05$. **(Dii)** Frequency change violin plots, stats indicate linear mixed effects model results and contrasts, *** $p < 0.001$, ** $p < 0.01$, * $p < 0.05$, † $p < 0.1$, Wald tests. **(E–H)** Show the same analyses/results as **A–D**, but for experiment 2. ROI, region of interest.

including Pz and surrounding electrodes, all from the hd-EEG system; see Methods), but only for the post-peak and post-trough, where alpha frequencies between ~9 to 10 and ~10 to 12 Hz were reduced, respectively (**Fig 2G**). However, we did not detect a change in frequency for Pz

stimulation (**Fig 2H** and **Fig 2G** in **S1 File**). Overall, these results demonstrate that the effects of αCLAS on alpha power and frequency are dependent on both the phase at which sounds are delivered and the target location. Alongside changes in power and frequency in experiment 1, we also observed phase-specific changes in connectivity that were specific to the alpha band and to αCLAS targeting of the Fz electrode in experiment 1 (**Figs I, J,** and **K** in **S1 File**).

## Single-pulse αCLAS

**Auditory evoked potentials provide a putative phase-reset mechanism by which αCLAS may operate.** To understand why sounds administered at different phases of the alpha oscillation appear to speed it up or slow it down in a spatially specific manner, we conducted 2 new experiments, similar to the previous ones, but with the critical difference that sounds were administered as isolated/single-pulse stimuli. For these 2 new experiments, we targeted again the alpha rhythm at Fz (where we previously showed an effect of αCLAS on alpha activity; experiment 1) and Pz (where we did not observe an effect of αCLAS; experiment 2), respectively (experiment 3: $N = 8$; experiment 4: $N = 7$). Participants sat with their eyes closed and listened to single pulses of sound (196 per phase condition), locked to the same 4 phases as before, and interleaved with silent periods (**Fig 1B**).

To investigate the changes in frequency observed previously, we adopted a phase-reset model. In a phase-reset model, the phase (and hence, the frequency) of an oscillation changes abruptly following reception of a stimulus. Specifically, this model predicts that the extent and trajectory (i.e., faster or slower) of the reset is dependent on the phase at the onset of perturbation, consistent with our findings from experiment 1. For example, if an oscillation reliably resets to a particular phase with a fixed latency following a stimulus, the trajectory of its phase (and consequently, the frequency of the oscillation) would differ depending on the phase at stimulus onset—this can take 2 forms: an advancing of the cycle or a delaying, to meet the reset [37].

In experiments 3 and 4, we observe an auditory-evoked potential (AEP) in the broadband signal following sound stimulation to each phase condition with stereotypical response, more pronounced in central and frontal than parietal brain areas, e.g., [38,39] (**Fig 3Ai** and **3Ci,** show the AEP at Fz and Pz from the hd-EEG system, respectively). The trajectory of both the amplitude (**Fig 3Aii** and **3Cii**) and phase (**Figs 3Aiii** and **5Ciii**) components of alpha (i.e., amplitude and phase) were computed using the ecHT algorithm offline at the targeted locations (Fz, experiment 3 and Pz, experiment 4, both using the hd-EEG system). Before sound onset (at time 0) the 4 αCLAS phase conditions showed, as expected, orthogonal time course profiles (ANOVA shows a main effect of αCLAS phase on signal amplitude for the broadband, **Fig 3Ai** and **3Ci**, and alpha band filtered signal, **Fig 3Aii** and **3Cii**). The pre-stimulus amplitude was higher at Pz (**Fig 3Ci**) in agreement with the more pronounced distribution of alpha in occipito-parietal areas at rest with eyes closed [1] (mean pre-stimulus log alpha band power ± std, experiment 3: −6.30 ± 0.38; experiment 4: −5.58 ± 0.60; two-sample $t$ test: $t(14) = −2.67$, $p = 0.018$). The post-stimulus time course response shows a high overlap between conditions at Fz (with consequent decrease in orthogonality and no significant effect of αCLAS phase on signal amplitude), while at Pz there remained significant differences between phase conditions following stimulus onset. The analysis of the phase time course using the Rayleigh test at each time point, confirmed that the conditions became significantly aligned at Fz in experiment 3, about 200 ms after stimulus onset (**Fig 3Aiii**) but not in Pz experiment 4 (**Fig 3Ciii**).

To address the effect of oscillation strength on perturbability, we sorted the AEPs into 8 octiles based on pre-stimulus alpha amplitude (see **Fig L** in **S1 File**).The pre-stimulus alpha

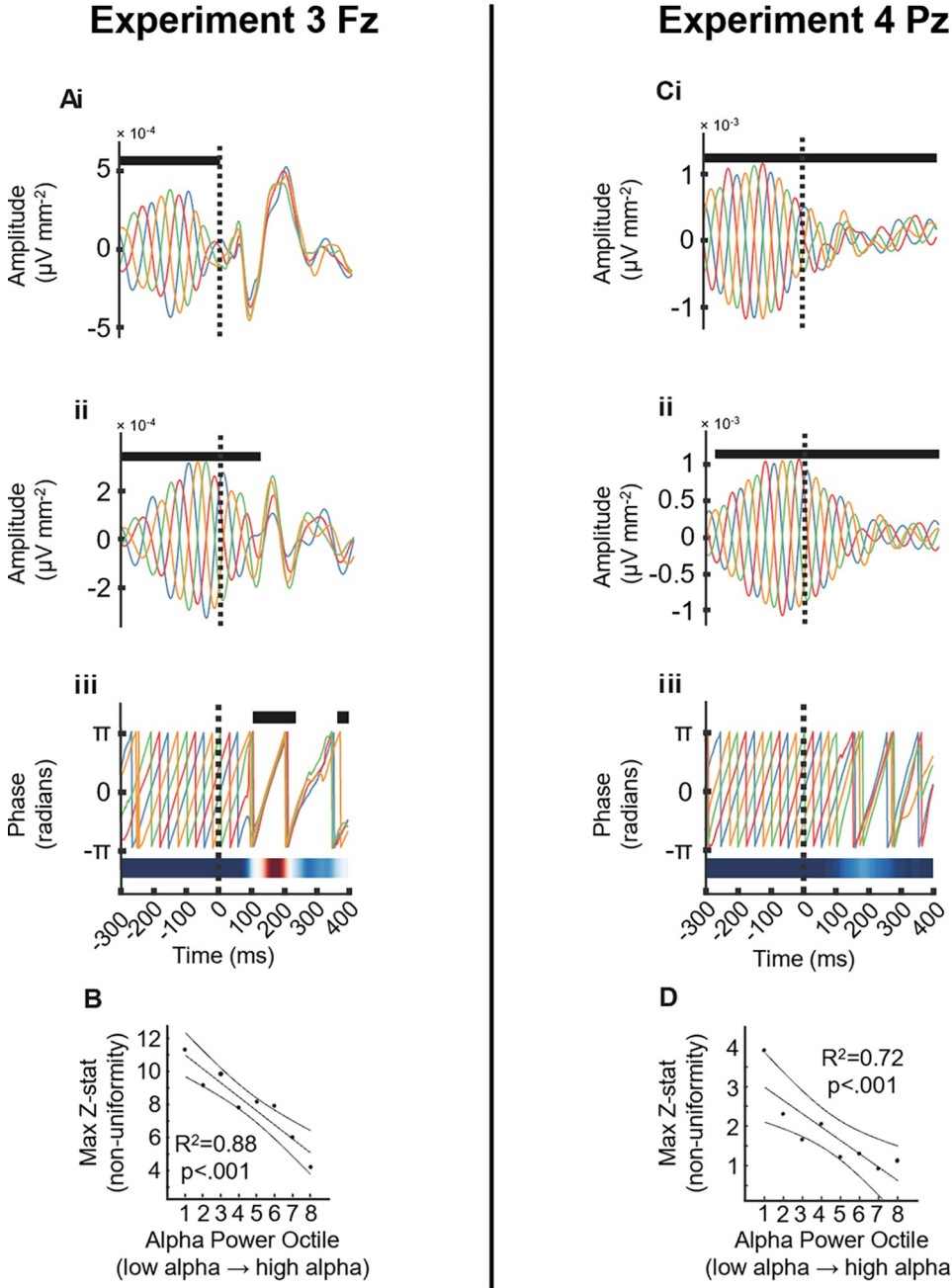

**Fig 3. AEP and phase reset.** (**A** and **C**) AEP at Fz in experiment 3 (**A**) and Pz in experiment 4 (**C**) for (**i**). Broadband EEG signal (1–40 Hz) AEP*. (**ii**) Amplitude component of alpha band (7.5–12.5 Hz) endpoint-corrected Hilbert transformed AEP*. (**iii**) Instantaneous phase of alpha band (7.5–12.5 Hz) from endpoint-corrected Hilbert transformed AEP**. (**B** and **D**) Pre-stimulus alpha power vs. maximum post-stimulus Z-statistic (resulting from Rayleigh test, and demonstrating significant unimodality, i.e., phase synchrony across conditions) at Fz in experiment 3 (**B**) and Pz in experiment 4 (**D**), line, confidence intervals, and statistics from linear regression. Fz and Pz electrodes from the hd-EEG system. * Black marks indicate ANOVA $p < 0.05$. ** Black marks indicate Rayleigh test $p < 0.05$, heatmap shows time series of Z-statistic. AEP, auditory-evoked potential.

amplitude showed a strong linear relationship with reset magnitude at both Fz and Pz (**Fig 3B** and **3D**), indicating that the lower the alpha amplitude (octile 1) the stronger the phase reset.

## Further characterisation of the phase response to alpha phase-locked sound perturbations

A phase-reset is often explored using phase transfer curves (PTCs) [37,40–42], in which the starting phase (i.e., at stimulus onset) is plotted against end phase (i.e., phase after a certain amount of time has expired). Phase response curves (PRCs) provide similar information, but instead of the end phase, the phase change is plotted. These curves allow both the magnitude and the specific nature of the reset to be summarised, detailing which start phases lead to an advance or delay of the cycle, and to what extent.

The PRCs and PTCs confirmed that sounds phase-locked at Fz, in experiment 3 (**Fig 4A–4C**), induced a greater reset by 200 ms than Pz in experiment 4 (**Fig 4D and 4E**), although there may still be a weak reset at Pz (see **Fig L** in **S1 File** for phase reset in octile 1 where alpha power is lower). Oscillator theory would distinguish these 2 types of resetting as type 0 and type 1, respectively—type 0 describes a strong input/reset, characterised by a phase response curve with a gradient of 0 (as in Fz, **Fig 4A**), while type 1 describes a weak input/reset, typified by a phase response curve with a gradient of 1 (as in Pz, **Fig 4D**) [37,40–42]. Additionally, when targeting Fz, the phase-dependent slowing and quickening observed in the PRC, agrees

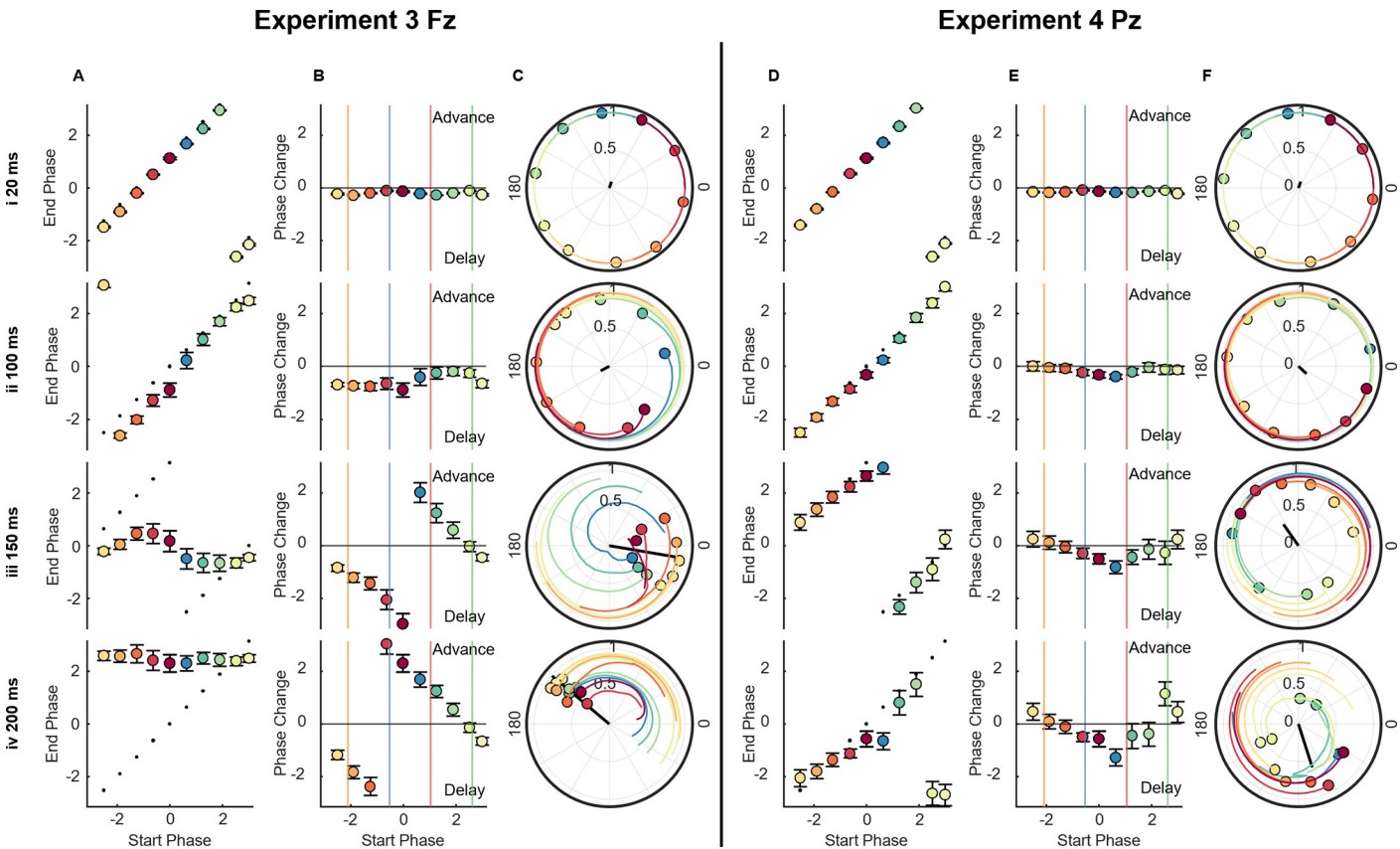

**Fig 4. Phase transfer and phase response curves.** (**A** and **D**) PTC, showing start phase vs. end phase (in radians) for Fz in experiment 3 (**A**) and Pz in experiment 4 (**D**). Dashed line indicates predicted phases, based on a 10 Hz oscillation. Error bars indicate standard error of the mean, computed in a circular fashion. (**B** and **E**) PRC, showing start phase vs. phase change from predicted phase (in radians) for Fz in experiment 3 (**B**) and Pz in experiment 4 (**E**). Horizontal line indicates zero. Vertical coloured lines indicate where actual conditions fell in their stimulus onset phase. Error bars indicate standard error of the mean, computed in a circular fashion. (**C** and **F**) Group phase and resultant for each of the 10 starting phase bins. Black line indicates resultant of all phase bins. Coloured lines indicate trajectory of each phase bin, across the preceding 50 ms, for Fz in experiment 3 (**C**) and Pz in experiment 4 (**F**). All figures are shown for delays 20 ms, 100 ms, 150 ms, and 200 ms. Fz and Pz electrodes from the hd-EEG system. PRC, phase response curve; PTC, phase transfer curve.

with what was seen in experiment 1; post-peak resulted in a quickening and post-trough achieved the opposite, but the remaining 2 conditions show more ambiguous responses. This ambiguity may be explained by the complexity associated with repeated perturbations. If an oscillator phase-resets with a fixed latency following a perturbation, but its frequency is allowed to vary, the phase-response profile should be frequency dependent—thus, if one pulse brings about a change in frequency, a second pulse may interact with the oscillation differently.

### αCLAS during the awake-to-sleep transition modulated alpha frequency in a phase-dependent manner

The phase-dependent effects observed in the previous experiments suggest that phase may be leveraged using αCLAS to modulate explicit brain processes that rely on dynamics associated with alpha oscillations. As the brain transitions away from wakefulness during the process of falling asleep, alpha oscillations undergo a dramatic disappearance across the cortex. This process first systematically described in the 1930's [43,44] is still employed to define the first stage of NREM sleep. If our αCLAS modulation is functionally meaningful, then it might also modulate alpha-associated processes in a phase-dependent way.

To investigate this, 16 participants took part in a within-subjects, sham-controlled, cross-over nap study (experiment 5), in which repetitive-αCLAS was once more employed, phase-locked to alpha at Fz, where we previously demonstrated an effect of αCLAS. In separate visits, participants were exposed to 3 conditions (sham, i.e., no sound, pre-peak repetitive-αCLAS and pre-trough repetitive-αCLAS), and given a 31-min opportunity to sleep, the first minute served as baseline, after which sounds played for 15 min (in the stimulation conditions), followed by 15 min of silence (**Fig 1B**). This design allowed us to explore online (i.e., during) and offline (i.e., post-stimulation) effects of phase-locked alpha stimulation on metrics of sleep and brain dynamics.

Comparable to the results of experiment 1 (**Fig 2Bii**), we observed differences between the stimulation conditions in the power at Fz, which again predominantly presented as differences in power within the alpha band, and primarily saw a greater power at around 8 Hz in pre-trough stimulation (**Fig 5A**). We focussed on power during the first 5 min of stimulation here, during which the majority of participants remained awake as in experiment 1 (the analysis including the full duration of the nap opportunity can be seen in **Fig O** in **S1 File**).

We observed a striking and sustained difference in alpha frequency at the phase-locking electrode (i.e., Fz) between conditions during stimulation (i.e., initial 15 min), which ceased when stimulation ended (**Fig 5Bi**). The increase in alpha frequency during pre-peak compared to pre-trough condition was observed for 15 out of 16 participants (**Fig 5Bii**) and both conditions were significantly different in frequency to sham. Once stimulation ceased, so did the differences in frequency (**Fig 5Biii**). Furthermore, the distribution of ISIs differed between vigilance states further suggesting that the ISI and therefore the closed-loop algorithm used is dependent on the brain's physiology (see **Fig Q** in **S1 File**).

### αCLAS impacts sleep *macro*structure in an alpha phase-dependent manner

Sleep scoring showed the expected trajectory during the nap; all participants began in an awake state and the majority were asleep (defined by any stage of NREM) after approximately 10 min (**Fig 6Ai**). We observed some awakenings once stimulation ceased (dotted line in the time series plots in **Fig 6**) with some participants reported being asleep until the sounds stopped. Although unintended, this is consistent with the finding that the sleeping brain remains sensitive to changes in environmental stimuli [45]. Aside from these awakenings, all 3

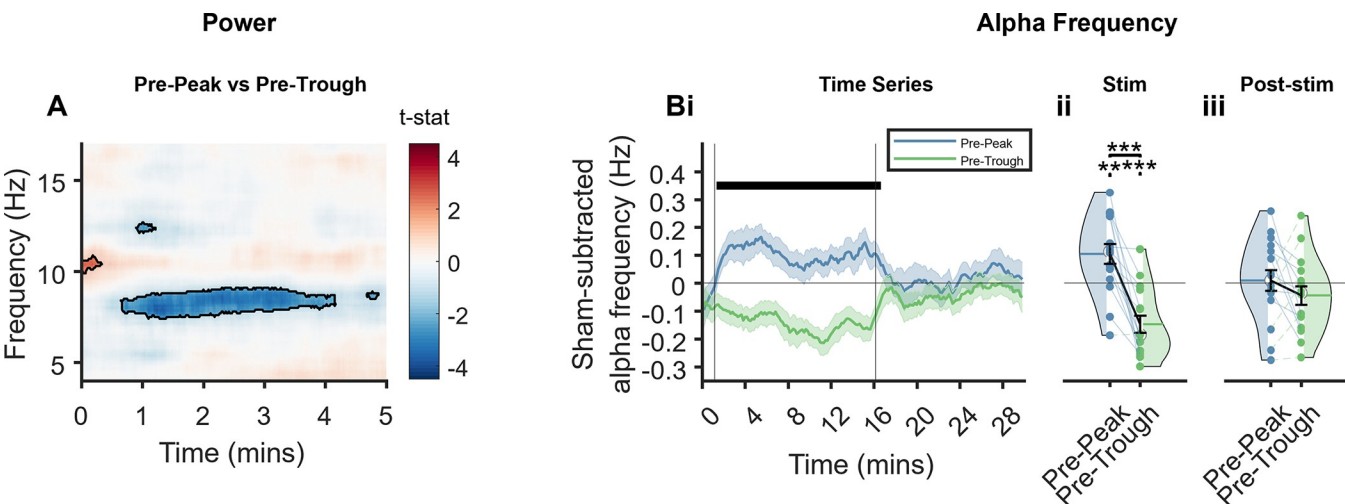

**Fig 5. Effects of stimulation phase on power and frequency at Fz. (A)** Time-frequency representation of differences between stimulation conditions during the first 5 min of stimulation, as per paired *t* tests. Red colours indicate pre-peak>pre-trough and blue indicates pre-peak<pre-trough. Black lines outline statistically significant clusters. **(B)** Sham-subtracted alpha frequency **(i)** across time, error bars are SEM vertical lines represent stimulation start and stop; thick horizontal bar indicates time points at which $p \leq 0.5$, as calculated by *t* test **(ii)** collapsed across the stimulation period; **(iii)** collapsed across the post-stimulation period. Fz electrode from the αCLAS EEG system as this electrode was not part of the PSG system. αCLAS, Alpha Closed-Loop Auditory Stimulation.

conditions showed a comparable trajectory and total sleep time (TST) did not significantly differ between conditions during either the stimulation or post-stimulation periods (**Fig 6A**).

Average sleep stage increased over time and was significantly different between conditions—pre-trough stimulation showed diminished sleep depth during stimulation (**Fig 6B**). When broken down further into N1 and N2+ (epochs labelled as N2/N3), these differences became more apparent with a diminished trajectory of cumulative N2+ sleep in the pre-trough condition, amounting to significantly less N2+ sleep during the stimulation period when compared to pre-peak and sham (**Fig 6C**; for the trajectory of N1 sleep, see **Fig R** in **S1 File**).

Latency to onset of the first epoch of sleep (any stage) did not differ between conditions (**Fig 6E**). However, participants took significantly longer to reach the first epoch of N2+ in pre-trough stimulation (**Fig 6G**). These results show that pre-trough stimulation impacted sleep macrostructure, relative to sham, while pre-peak stimulation did not.

## αCLAS impacts sleep *micro*structure in an alpha phase-dependent manner

Sleep scoring is a useful way to broadly classify sleep physiology, but its drawbacks are well known. Namely, the brain probably exists on something of a continuum rather than conforming to 4 or 5 discrete states. To arrive at a finer grain measure of EEG, we characterised both rhythmic, i.e., oscillatory, and arrhythmic, i.e., aperiodic or 1/f, EEG activity using eBOSC [46] on the data obtained throughout the nap opportunity (to verify the utility of eBOSC features in distinguishing sleep stages, see **Figs S** and **T** in **S1 File** for trajectory across time of all eBOSC EEG features).

The majority of these metrics confirmed findings from the sleep scoring (**Fig 6** showing eBOSC features for frontal electrodes; all regions with a main effect of condition are shown in **Fig U** and **Tables E** and **F** in **S1 File** contain full statistics). Specifically, during pre-trough αCLAS features which increase in magnitude during the natural progression of sleep were reduced, namely offset of 1/f slope (**Fig 6J**) and sigma abundance (representative of spindles, **Fig 6I**), while features which decrease with sleep were greater, i.e., exponent of 1/f slope

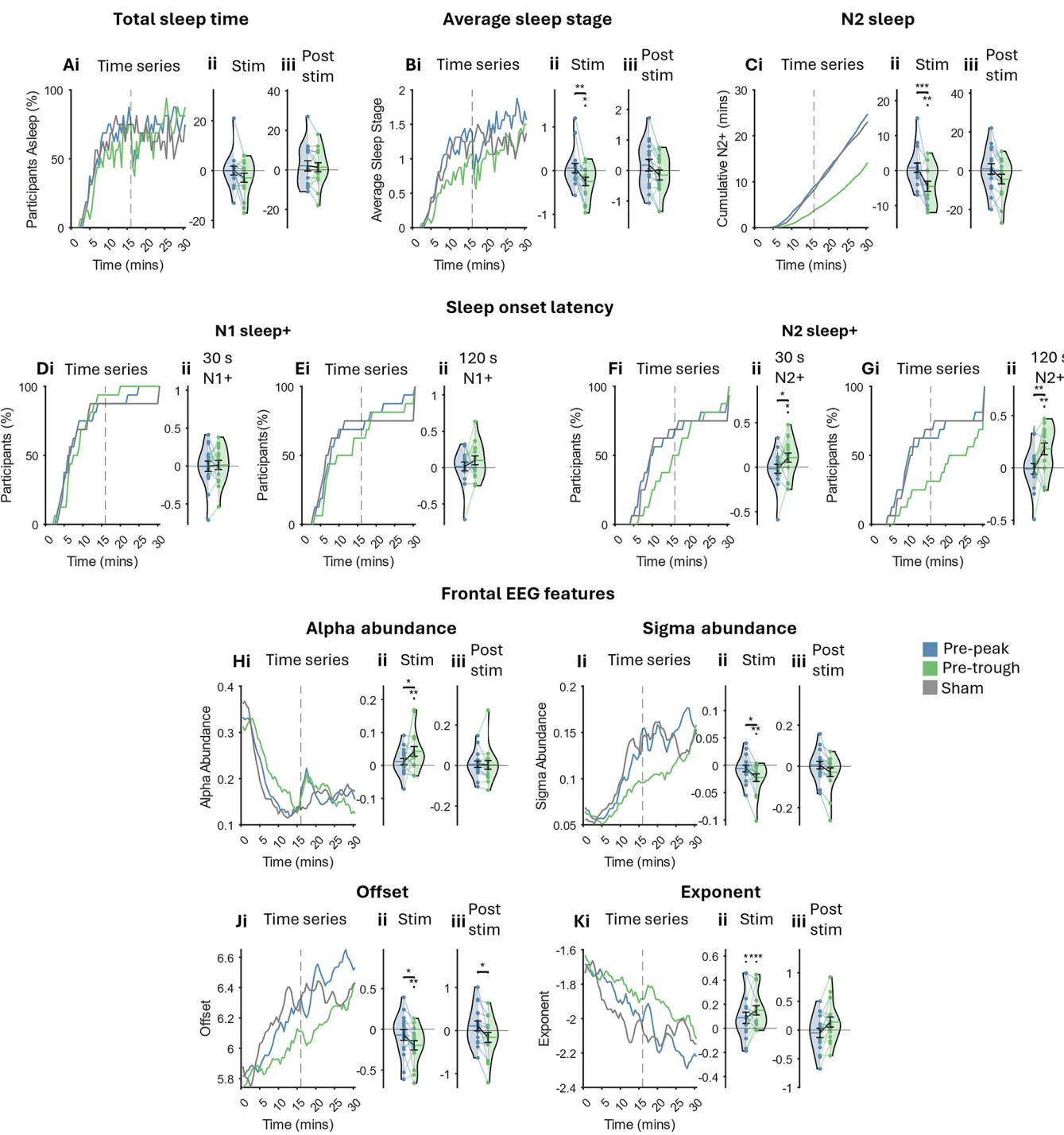

**Fig 6. Sleep structure and EEG features.** **(A)** **(i)** Time series of the percentage of participants in any stage of vigilance (each state was given a numerical value: awake– 0, N1/NREM1–1, N2/NREM2–2, N3/NREM3–3, and no REM sleep was observed). Dashed line indicates end of stimulation. **(ii)** Sham-subtracted TST collapsed across stimulation period. **(iii)** Sham-subtracted TST collapsed across post-stimulation period. **(B)** **(i)** Average sleep stage time series. Dashed line indicates end of stimulation. **(ii)** Sham-subtracted average sleep stage collapsed across stimulation period. **(iii)** Sham-subtracted average sleep stage collapsed across post-stimulation period. **(C)** **(i)** Cumulative N2+ (i.e., N2/N3) sleep time series. Dashed line indicates end of stimulation. **(ii)** Sham-subtracted time in N2+ sleep collapsed across stimulation period. **(iii)** Sham-subtracted time in N2+ sleep collapsed across post-stimulation period. **(D)** **(i)** Time series of participants (%) having had at least 30 s of sleep (N1+; any stage). **(ii)** Sham-subtracted log-transformed latency of sleep onset (any stage, at least 30 s). **(E)** **(i)** Time series of participants (%) having had at least 120 persistent seconds of sleep. **(ii)** Sham-subtracted log-transformed latency to the onset of the first persistent 120 s of sleep (any stage). **(F)** **(i)** Time series of participants (%) having had at least 30 s of N2+ sleep. **(ii)** Sham-subtracted log-transformed latency of N2+ sleep onset. **(G)** **(i)** Time series of participants (%) having had at least 120 persistent seconds of N2+ sleep. **(ii)** Sham-subtracted log-transformed latency to

the onset of the first persistent 120 s of N2+ sleep. **(H–K)** Time series of EEG features (left) averaged across channels F3 and F4 in the PSG system and averaged across participants. Sham-subtracted features collapsed across stimulation period (middle) and post-stimulation period (right). Note: statistics were run prior to sham subtraction. Post hoc comparisons were only carried out when a main effect of condition was seen in LME. Significance bars between violins indicate a difference between stimulation conditions, significance marks over violins indicate a difference from sham. *** $p < 0.001$, ** $p < 0.01$, * $p < 0.05$, † $p < 0.1$, $t$ tests. Each violin shows a dot per participant, per condition, horizontal lines indicate the mean. NREM, non-rapid eye movement; TST, total sleep time.

(Fig 6K) and alpha abundance (representative of rhythmic alpha activity, Fig 6H). Of note, the exponent was significantly shallower than sham during both types of stimulation, but this was the only metric by which sham and pre-peak αCLAS could be distinguished.

In sum, the trajectory of sleep was modified by αCLAS in a phase-dependent manner. Specifically, while the onset and duration of sleep did not differ between conditions, its composition did; a "shallower" sleep was observed when sounds were repeatedly administered just prior to the troughs of alpha oscillations, relative to pre-peak locked sounds and the absence of sounds.

## Discussion

Across a series of independent experiments we have shown: (i) αCLAS can induce phase-dependent modulations of alpha activity in a spatially specific manner (experiments 1 and 2); (ii) this effect is dependent on oscillation amplitude (experiments 3 and 4); (iii) sounds delivered at different phases influence the trajectory of alpha oscillations (experiments 3 and 4); and (iv) the transition to sleep can be modulated by αCLAS (experiment 5).

We first demonstrated that phase-dependent responses to auditory stimulation of the alpha oscillation can be elicited in the resting brain. By employing hd-EEG and targeting 2 distal cortical locations with naturally different alpha power, we observed that repetitive-αCLAS modulated the alpha rhythm in a spatially specific manner, with changes to alpha power, frequency, and connectivity present when targeting the frontal, but not the parietal location. While the frequency changes may appear very small they are consistent with, or greater than, those derived from the same analysis method elsewhere in the literature [7,36,47].

Previous studies on phase-dependencies of neural oscillations consider phase an index of "cortical excitability," i.e., a cyclical measure of the cortex's sensitivity to stimuli. These studies focused primarily on secondary, or higher order, outcomes (perception, for example [5,7,8,48–51]) of stimuli delivered at a particular phase without making explicit predictions on the oscillation's response, or they expected a straightforward change in the amplitude of the targeted oscillation [52], akin to observations from sound stimulation to the slow oscillations of sleep [28–30]. While our stimulation effects agreed with the literature, insofar as they are phase-dependent, we found they could be better explained by changes to the frequency of oscillations, rather than amplitude.

By targeting the alpha rhythm at multiple phases while delivering single-pulse αCLAS, we tested whether a phase-reset mechanism would provide a parsimonious explanation to the observed alpha frequency modulation. Our results support this hypothesis and were consistent with an oscillator model, in which a perturbation will advance or delay an oscillation to varying degrees depending on the phase at which it is administered [37,41,42]. There is no unanimous consensus on how related or unrelated evoked potentials are to phase reset of oscillations [53] with ongoing attempts to adjudicate between the best models to explain the generation of evoked responses from EEG data [54–68]. Current evidence suggests that the differentiation between these models might not be feasible if based solely on non-invasive scalp EEG recordings [56,69]. Nevertheless, our results cannot simply be explained by the response evoked by the auditory stimulus, as we observed similar auditory evoked potentials across

phase conditions. In addition, our results showed that the frequency dependency on phase was maintained, for short (experiment 1) and sustained periods of stimulation (experiment 5).

Despite, phase-locking with high spatial selectivity, modulation of alpha activity was limited to phases targeted at the frontal region. We considered 2 complementary explanations for this phenomenon which are congruent with our findings. The first lies in the endogenous regional differences in alpha amplitude, i.e., higher in parietal than frontal locations. Indeed, our results demonstrate that a parietal phase-reset could only be observed in instances of low alpha power, and that alpha power and phase reset magnitude are strongly related. This is also in line with oscillator theory, which proposes a phase reset is dependent on the magnitude of both the oscillator and the perturbation (in our case, the magnitude of the perturbation, i.e., sound volume, is kept constant between experiments, but the oscillator amplitude varies from frontal to parietal regions), and agrees with prominent theories that the phasic inhibition exerted by alpha oscillations is proportional to their amplitude [15,70]. Secondly, the differential in the magnitude of AEPs at frontal and parietal regions could be a contributory factor to regional differences, depending on how related the generation of evoked potentials is to the phase reset of oscillations which is currently unresolved [54–68]. Future studies employing visual stimuli phase-locked to parietal alpha oscillations could help disentangle the contribution of evoked potential magnitude and oscillation amplitude in phase-dependent responses, since visual evoked potentials are larger over the visual cortex in spite of higher amplitude oscillations.

Finally, in addition to the immediate electrophysiological consequence of αCLAS, we have shown the functional significance of this approach, during the transition to sleep. Auditory stimuli modulated alpha frequency, once again, and sleep depth on a phase-dependent basis. We, put forward 2 hypotheses to explain these findings. In experiments 1 and 5, we showed that pre-peak αCLAS predominantly caused a decrease in alpha power, while pre-trough αCLAS did the opposite, i.e., an increase in alpha power. Similarly, in experiment 5, we show that the abundance of alpha oscillations was greater in pre-trough stimulation than in either pre-peak or sham stimulation, despite these conditions not differing in their quantity of wake epochs. Accordingly, we suggest pre-trough αCLAS may have impeded the disappearance of alpha, preventing sleep from progressing further, while pre-peak αCLAS did not. Alternatively, differences in sleep depth may simply reflect phasic differences in sensitivity to sound. In this scenario, pre-trough locked sounds arrived at a more suitable phase at which to perturb the brain and in consequence the sleep process, in a similar vein to the cyclic excitability put forward in the visual neuroscience literature [5,6,8,14,15,48,49,71]. Studies using intracranial electroencephalographic (iEEG) recordings in humans or animal models could help to elucidate the direct neural response and offer further mechanistic insights.

A general limitation of our studies is that our closed-loop stimulation did not account for variations in the endogenous alpha power. This was mitigated by the fact that participants had their eyes closed, a state of abundant alpha oscillations, and variations in alpha power were used to our advantage in experiments 3 and 4. However, in experiment 5, we see that ISIs deviate to other frequencies when alpha oscillations are reduced as participants fall asleep. While we cannot completely rule out an effect of ISI, or stimulation frequency in experiment 5, such that the impact of sound on sleep is only phase-dependent in the sense that the frequency of sound was phase-dependent, the differences in ISI (0.4 Hz on average) are very slight, rendering this explanation unlikely. Future studies can implement closed-loop strategies that include a detection threshold, thus limiting stimulation to periods when the target rhythm is most prominent. Albeit the presence of a genuine alpha oscillation does not seem to be a requisite to induce changes in alpha frequency, as observed in experiment 5 where alpha oscillations are naturally reduced as participants fell asleep. Another helpful addition may be the inclusion of a

random-sound condition—in experiment 5 we were unable to say for certain whether the pre-peak represented a moment of decreased sensitivity to sound, or pre-trough a window of increased sensitivity. Additionally, as data for some experiments were acquired in parallel, the 2 phases targeted in experiment 5 may not have been optimal to differentially affect the sleep onset process, owing to the similarities we observed between these conditions in experiment 1. Nevertheless, the relative consistency between findings across independent samples and experimental designs speaks to the general robustness of the findings and the presence of a phase-dependent effect. Finally, we used a mastoid reference for real-time phase-locking. The use of an on-line Laplacian reference, as seen in some other closed-loop experiments [22–24], may afford stronger, and further localised effects.

Alpha oscillations are involved in a number of fundamental processes. Here, we demonstrate that sound can be used to affect phase-dependent modulation of their activity. Future research could extend our findings into potential clinical applications. For example, by countering the decline in alpha frequency observed in older age and cognitive decline/dementia [72,73] or facilitating the increase in alpha frequency observed during increases in cognitive load [74]. There remains much to be explored, regarding the application of αCLAS to neural oscillation-dependent behaviours, but we propose that sound can provide more utility in this context than previously considered, particularly if stimulation is used to address the brain on its own terms, through closed-loop approaches.

## Materials and methods

### Ethics statement

All experiments were approved by the University of Surrey Ethics Committee (approval numbers: FHMS 19–20 103EGA and FHMS 22–23 123EGA) and conducted in accordance with the Declaration of Helsinki. All participants gave written informed consent and were compensated for their time.

### Participants

Experiments 1 and 2: 24 participants took part in experiment 1 and 31 in experiment 2. One participant was excluded from experiment 1 due to an error in the phase-locking. Three participants were excluded from experiment 2 due to missing triggers. This left a sample of 23 participants in experiment 1 (16 female, mean ± std age = 25.70 ± 5.46 years) and 28 participants in experiment 2 (18 female, mean ± std age = 20.77 ± 2.49 years). Two participants overlapped between the 2 experiments.

Experiments 3 and 4: 8 participants took part in experiment 3 (7 female, mean ± std age = 24.50 ± 5.90 years) and 7 in experiment 4 (6 female, mean ± std age = 26.71 ± 5.82 years). Five participants took part in both experiments. All participants were included in the analysis.

Experiment 5: 25 participants took part in this experiment, 9 of whom were excluded (1 for sleep-onset REM, 2 for technical issues, 5 for missing visits due to illness or other reasons, 1 for night shift between visits). The final sample was composed of 16 participants (14 female, mean ± std age = 22.38 ± 2.74 years).

No statistical methods were used to predetermine sample sizes, but our sample sizes are similar to those reported in previous publications investigating the effects of sound stimulation on physiological responses, which typically vary between 7 to 10 participants for studies measuring evoked responses as in experiments 3 and 4 [64,65,75,76] to about 19 for studies using closed-loop auditory stimulation, typically targeting slow-wave sleep activity [77]. Across experiments, all participants self-reported no history of neurological or psychiatric illness,

normal hearing, and a body mass index of less than 30. Participants taking part in experiment 5 were asked not to consume caffeinated drinks on the morning of the experiment and refrain from drinking alcohol the nights before the experiment. Participants gave written informed consent. The study conforms to the Declaration of Helsinki and received ethical approval from the University of Surrey's ethics committee. Data collection for different experiments occurred in parallel.

## Experimental design

Experiments 1 and 2: Participants sat in a comfortable armchair in a soundproof room with constant LED light (approx. 800 Lux). Participants were exposed to 8 runs (2 runs per phase) of auditory stimulation phase-locked to either Fz (experiment 1) or Pz (experiment 2). Each run comprised 5 blocks of 30 s stimulation, interleaved with 10 s of silence, and started with 10 s of silence. The order of the runs was counterbalanced across participants using a balanced Latin square. Each run lasted 200 s. At the beginning of each run, participants were asked to close their eyes and relax. At the end of each run, participants were told to open their eyes and electrode impedance was checked and, if necessary, adjusted. This procedure helped to avoid participants falling asleep during the session. In total, each session lasted approximately 2 h.

Experiments 3 and 4: Experiments were conducted under the same laboratory conditions as experiments 1 and 2. However, instead of blocks of continuous sound stimulation, participants were exposed to isolated bursts of sound. Participants were exposed to 8 runs (2 per phase targeted) of auditory stimulation phase-locked to either Fz (experiment 3) or Pz (experiment 4). Each run comprised 96 pulses of pink-noise, interleaved with between 1.5 and 5.5 s of silence (randomly generated, uniformly distributed, mean ± std = 3.5 ± 0.88 s). The order of the runs was counterbalanced across participants using a balanced Latin square. Each run lasted around 350 s. Total number of pulses per condition was 192, and the total duration of the session was around 2 h.

Experiment 5: Participants attended the Surrey Sleep Research Centre lab for a total of 4 sessions, the first of which was an adaptation visit, to allow participants to acclimatise to the sleep centre and protocol. Visits were separated by at least 2 days. Each session had an approximate duration of 3 h and in addition to the nap included a memory task, Karolinska drowsiness test (KDT) and Karolinska sleepiness scale (KSS), which were not analysed here. Throughout the study, starting at the adaptation visit, participants were requested to use a tri-axial GENEActiv accelerometer (Activinsights, Kimbolton, United Kingdom), on their non-dominant wrist, and complete sleep diaries, which were checked to confirm a regular sleep schedule. Participants also completed a Pittsburgh Sleep Quality Index (PSQI) questionnaire to assess sleep health.

During the nap portion of the visit, participants lay supine in bed in a temperature-controlled, sound-attenuated, pitch-dark, windowless room for the duration of the nap opportunity (i.e., 31 min). Participants were asked to keep their eyes closed for the entire nap period, regardless of whether they were asleep or not. All visits took place in the morning, starting at 07:00, 08:00, or 09:00 h, with the naps starting approximately 2 h into the visit. Visits were scheduled in the morning as sleep latencies vary during the day, with longer latencies observed in the morning compared to afternoon in young adults [78]. Thus, providing a possible longer window in which to study the effects of αCLAS on the dynamics of falling asleep. The time of each visit, and hence each nap, was consistent across visits for each participant. Participants were blinded to the condition, as was the researcher providing instructions to the participant.

## Phase-locked auditory stimulation

Real-time closed-loop phase-locked auditory stimulation was delivered using a custom-made device as described in [35]. This device utilised 3 electrodes; a signal electrode (placed at Fz or Pz, depending on the experiment), a reference, and ground (both placed at the right mastoid). Shielded electrodes (8 mm, BIOPAC Systems Inc.) were used for the signal and reference, and an unshielded electrode was used for the ground (8 mm, BIOPAC Systems Inc.). The signal was filtered using a 7.5 to 12.5 Hz bandpass, and the phase of the resultant signal was computed in real-time using the ecHT. The device used for experiments 1 to 4 operated at a sampling rate of 500 Hz, while that used in experiment 5 operated at 826 Hz.

Across experiments, sounds were 80 dB, 20 ms pulses of pink noise, delivered to the participants via earphones (Etymotic model ER-2 in experiments 1 to 4 and Shure model SE215-CL-EFS in experiment 5). During pilot testing with continuous stimulation over a period of minutes, the volume of the sound did not prevent drowsiness and was therefore also deemed appropriate for experiment 5. Sounds were phase-locked to 4 orthogonal phases (i.e., providing maximum separation between each phase condition) of alpha in experiments 1 to 4 (60˚, 150˚, 240˚, and 330˚) and 2 phases in experiment 5 (150˚ and 330˚). In the sham condition in experiment 5, phase-locking was still computed, and markers were logged for all "stimuli" (dummy targeting 330˚), but the volume was set to 0 dB ensuring that no sounds were played to the participant.

## EEG data acquisition and sleep monitoring

Experiments 1–4: High-density (hd)-EEG recordings were acquired in parallel with the phase-locking EEG device (i.e., αCLAS EEG system). Hd-EEG recordings were obtained using an actiCHamp-Plus amplifier (Brain Products GmbH) and 128-channel water-based R-NET electrode cap (EasyCap GmbH, Brain Products GmbH) positioned according to the international 10 to 20 positioning system. Electrode impedances were below 50 kΩ at the start of each recording, as per the manufacturer's guideline, and were adjusted between each run, to ensure accurate impedances throughout the recording. The hd-EEG electrodes matching those used for phase-locking in the αCLAS EEG system (i.e., Fz or Pz) were moved to be adjacent to those in the αCLAS EEG system. All channels were recorded referenced to Cz, at a sampling rate of 500 Hz.

Experiment 5: Sleep recording equipment was acquired in parallel with the phase-locking αCLAS EEG system. Sleep recording equipment was a Somnomedics HD system with Domino software (v2.8), sampled at 256 Hz (S-Med, Redditch, UK) using the American Academy of Sleep Medicine (AASM) standard adult EEG montage. The system included 9-channel EEG placed according to the 10 to 20 system at F3, F4, C3, C4, O1, O2, left mastoid (A1), and right mastoid (A2), and referenced to Cz, as well as submental electromyography (EMG), electro-oculogram (EOG), and electrocardiogram (ECG).

The αCLAS EEG system was programmed to send triggers at the beginning of stimulation blocks in experiments 1 and 2, which were recorded in the BrainVision Recorder software (Brain Products GmbH) alongside EEG data, via a TriggerBox (Brain Products GmbH). In addition, in experiments 1 to 4, we used a StimTrak (Brain Products GmbH) that logged a trigger whenever a sound was played, allowing us to assess phase-locking in the hd-EEG data and to time-locked evoked potentials. In experiment 5, an analogue marker was sent from the αCLAS EEG system to the S-Med PSG system, so as to align EEG recordings according to the start of the nap.

## EEG preprocessing

All EEG data were preprocessed and analysed in MATLAB 2021a (Mathworks, Natick, Massachusetts, United States of America). EEG filtering was carried out using the FieldTrip toolbox [79], all other preprocessing was carried out using EEGLAB v2022.0 [80]. All topoplots and related statistics were produced using FieldTrip [79].

Hd-EEG data were high-pass filtered (1 Hz) and notch filtered (50 Hz, 100 Hz), before automatic subspace reconstruction (ASR) was used to remove noisy channels and artefacts. EEG data from experiment 5 were filtered between 1 and 30 Hz, before artefacts were manually identified and removed. The hd-EEG data from experiments 1 to 4 was subjected to a scalp current density (i.e., Laplacian) re-referencing, while the EEG data from experiment 5 remained referenced to Cz for all analyses except sleep scoring which was performed with data referenced to the linked mastoids.

## EEG analysis

Accuracy of phase-locking was calculated for each channel, participant, condition, and experiment using the following method. For every stimulus, the EEG was epoched in 1 s windows up to and including stimulus onset, before the ecHT algorithm was applied, and the final phase estimate in each window was taken. The ecHT algorithm minimises the Gibbs phenomenon by applying a causal bandpass filter to the frequency domain of an analytic signal, thus selectively removing the distortions to the end part of the signal. This strategy has been shown to provide accurate computations of the instantaneous phase and envelope amplitude of oscillatory signals [35]. The resultant vector length of these values was then computed, giving a value between 0 and 1, representing perfectly uniform and perfectly unimodal circular distributions, respectively. The accuracy of phase-locking for the group was summarised by computing the average resultant of all participants and computing a V-test. V-test was calculated using phase angle per participant and the respective target phase angle (i.e., 60˚, 150˚, 240˚, 330˚ depending on the condition).

For all analyses regarding power, the same method was used; power was computed in 1-s epochs, 2 to 30 Hz, in steps of 0.1 Hz, Hamming window, using the "pwelch.m" function. For all analyses regarding frequency, Cohen's frequency sliding method was used [36].

**Experiments 1 and 2.** Frontal and parietal regions of interest (ROI) were defined a priori and included 4 electrodes, the corresponding phase-locking electrode in the hd-EEG system (i.e., Fz or Pz) and the 3 surrounding electrodes. The frontal cluster comprised AFz, Fz, AFF1h, and AFF2h. Whist the parietal ROI consisted of Pz, POz, PPO1h, and PPO2h. ROIs only included electrodes from the hd-EEG system and chosen to balance spatial specificity (i.e., a small number of channels located close to the phase-locking channel) and sensitivity (i.e., the benefit of having data from more channels to improve signal-to-noise ratio to detect and effect), while keeping the number of channels per ROI constant across analyses and experiments.

Power spectra in **Fig 1C** were computed from the off-period using the αCLAS EEG system and were normalised by z-scoring, i.e., expressing power at each frequency as number of standard deviations from the mean power, per participant. This was so as to plot all participants together, without individual differences leading to large disparities between participants' power spectra. The actual power values here are of no consequence, since the aim was only to display a peak within the alpha band. Peaks were detected using the MATLAB function "findpeaks.m". If a peak was identified within the alpha band used by the αCLAS system (i.e., 7.5 to 12.5 Hz), it was marked on the plot.

For all subsequent analyses of power spectra in experiments 1 and 2, log-transformed ratios of "on" to "off" were computed for every frequency bin. Before running statistics and plotting time-frequency representations, data in experiments 1 and 2 were smoothed across time, using a moving mean window of 10 s. "movmean.m" function, with a window argument of "[10]", meaning each second in the plot was an average of the 5 preceding and 5 proceeding seconds. Statistics were computed for these time-frequency representations using either one-way ANOVA or $t$ tests as appropriate. Statistically significant ($p < 0.05$) contiguous clusters of 30 or more time-frequency points were kept and considered significant, and smaller clusters were discarded to control for multiple comparisons. IAF (for regression with inter-stimulus interval) was calculated at the respective phase-locking electrode (i.e., Fz/Pz from the αCLAS EEG system) using the frequency sliding method [36].

Connectivity was computed for 4 frequency bands (delta 1 to 4 Hz, theta 4 to 7 Hz, alpha 8 to 12 Hz, beta 13 to 30 Hz), for every second of clean EEG data. Two metrics were used, phase-locking value (PLV) and phase lag index (PLI). In both cases, the following was carried out for each channel pair: the data from each channel was filtered in the band of interest using a second-order Butterworth filter; phase difference between the 2 channels was estimated for all time points using Hilbert transform-derived phase estimates; for each second, the resultant vector length of phase differences was computed. The only difference between PLV and PLI was that in the case of PLI, phase differences of 0 mod π were discarded prior to computing the resultant. When comparing conditions, these measures of connectivity were averaged across the "on" (sound stimulation) period. When computing stimulation-induced changes to connectivity, these measures were separately averaged across the "on" and "off" periods before comparing the 2.

**Experiments 3 and 4.** Broadband auditory evoked potentials were computed by filtering the data from the hd-EEG system between 1 and 40 Hz (second-order Butterworth), and averaging over all trials, per condition, per participant. Narrowband (7.5 to 12.5 Hz) auditory evoked potentials (both amplitude and phase) were computed by applying the ecHT algorithm —a 1-s window was slid in increments of 1 data point (2 ms), each window was subjected to ecHT, and the estimate for the final time point in that window was assigned to that time point in the data.

To assess unimodality of phase angles between conditions (and hence, a phase reset), a Rayleigh test was carried out for each time point, using the average phase angle and resultant for each participant, for each condition. Time points at which the Rayleigh $p \leq 0.05$ were considered to have significant unimodality.

Trials were sorted into octiles on the basis of pre-stimulus (−200 to −100 ms) alpha power, (mean ± SD number of trials per octile per participant: experiment 3: 22.69 ± 0.998; experiment 4: 21.86 ± 2.37). This allowed us to assess the effect of oscillation amplitude on phase-reset. The aforementioned phase estimates were also used to sort all evoked potentials by stimulus onset phase, into ten 36° bins centred on 0°, 36°, 72°, 108°, 144°, 180°, 216°, 252°, 288°, and 324° (mean ± SD number of trials per phase bin per participant; experiment 3: 74.02 ± 1.45, minimum: 67.1, maximum: 80; experiment 4: 72.19 ± 2.72, minimum: 67.1, maximum: 80.4). This allowed phase response curves to be computed at various latencies, by plotting starting phase bin against end phase, and to derive phase transfer curves by plotting starting phase bin against change from expected phase (assuming a 10 Hz oscillation).

**Experiment 5.** Sleep was scored by an experienced sleep technician blinded to the experimental condition and unfamiliar with the details of the experimental protocol, according to AASM guidelines [81]. Scoring provided 30 s epochs labelled as wake, non-rapid eye movement (NREM or N) N1, N2, N3 or rapid-eye movement (REM) sleep. In order to better assess the dynamics of the ecHT's stimulation, we computed inter-stimulus intervals ISI's by taking

the difference in time between each stimulus and the stimulus that proceeded it, before converting this value to Hz. ISI's greater than 0.5 s (<2 Hz) were discarded before the average ISI was computed by taking the median. The proportion of ISI's which fell in canonical bands (theta 4 to 7 Hz, alpha 8 to 12 Hz, sigma 12 to 18 Hz) was then computed to demonstrate that the behaviour of the stimulator followed the oscillatory features of the sleep stages.

Power calculations, comparable to those in experiment 1 and 2, were computed in the same way, but here a window of 1 min was used for the moving mean owing to the much longer experiment (31 min).

We employed eBOSC [46] to estimate both rhythmic, i.e., oscillatory, and arrhythmic, i.e., aperiodic or 1/f, EEG activity (i.e., offset and exponent). For eBOSC analysis, for each channel, EEG data were divided into 30 s epochs, since, during a dynamic process such as sleep onset, a single fit and threshold for oscillation detection is unlikely to be optimal. For each of these epochs, eBOSC was run between 1 to 30 Hz, in steps of 0.25 Hz, excluded frequencies for background 1/f fit 7:16 Hz. For rhythmic activity, only oscillations with a duration of at least 3 cycles were analysed to estimate abundance (i.e., the total duration of rhythmic episodes as a proportion of the length of the analysed segment). The abundance computed was then averaged within canonical bands for each epoch. This was done for all channels (F3, F4, C3, C4, O1, O2) before further averaging into regions (frontal: F3, F4; central: C3, C4; occipital: O1, O2).

## Statistical analyses

Unless stated otherwise, statistics were run in the following manner. Cluster-corrected permutation $t$ tests or ANOVA were run on FieldTrip [79] and used for topographical representations. Linear mixed-effects models (LMEM), in which participant was a random effect, were used throughout. Where these LMEM registered a significant main effect ($p \leq 0.05$, as computed using the *anova* function) post hoc comparisons between all groups were carried out using the "emmeans" toolbox (https://github.com/jackatta/estimated-marginal-means), so as to control for multiple comparisons. Results for experiment 5 feature a number of "stim minus sham" plots—in these cases, an LMEM was run on the data from the 3 groups, prior to subtracting sham, and the sham-subtraction was applied for visualisation purposes. All circular statistics were conducted using the "circstat" toolbox [82].

## Supporting information

**S1 File. Fig A. Comparison of the resultant values using different referencing schemes.** The resultant was calculated in electrode Fz of the hd-EEG system in experiment 1 using 2 referencing schemes, Laplacian (as used for all analysis in the main manuscript) and right mastoid (to match the referencing scheme used in the αCLAS EEG system). Channel Fz in the hd-EEG system was adjacent to this channel in the αCLAS EEG system, and the mastoid channel in the hd-EEG system was taken from TP10. The resultant values using the right mastoid reference are higher and more similar to those reported for the αCLAS EEG system. Violin plots show the resultant for each targeted phase in experiment 1 in each reference scheme. Black lines represent each participant. Stats indicate output of paired samples $t$ test between the resultant for each reference scheme, *** indicates $p < 0.001$. **Table A. Data from closed-loop EEG device**. **Table B. Data from high-density EEG device**. **Fig B. Phase-locking accuracy across conditions**—topography of phase-locking accuracy (average resultant) across 4 conditions for experiments 1 and 2, targeting Fz and Pz electrodes, respectively (blue circles). White marks indicate channels at which resultant >0.4 and $p < 0.05$. Black marks indicate channels at which resultant <0.4 and $p < 0.05$; $p$-values from FDR-corrected z-test for non-uniformity.

**Fig C. Power change ANOVA for experiment 1** –**(A)** Topography of permutation ANOVA stats for frequencies across the spectrum. Each topoplot is computed using +/- 0.2 Hz around the labelled frequency. Colours and colourbars indicate F statistics. White marks indicate cluster-corrected $p$-values <0.05. **Fig D. Power change ANOVA for experiment 2** –**(A)** Topography of permutation ANOVA stats for frequencies across the spectrum. Each topoplot is computed using +/- 0.2 Hz around the labelled frequency. Colours and colourbars indicate F statistics. White marks indicate cluster-corrected $p$-values <0.05. **Fig E. Power change ANOVA for experiment 1** –**(A)** Topography of permutation ANOVA stats for various frequencies in and around the alpha band. Each topoplot is computed using +/- 0.2 Hz around the labelled frequency. Colours and colourbars indicate F statistics. White marks indicate cluster-corrected $p$-values <0.05. **Fig F. Power change ANOVA for experiment 2** –**(A)** Topography of permutation ANOVA stats for various frequencies in and around the alpha band. Each topoplot is computed using +/- 0.2 Hz around the labelled frequency. Colours and colourbars indicate F statistics. White marks indicate cluster-corrected $p$-values <0.05. **Fig G. Frequency estimates from off and on periods in experiments 1 and 2**. **(A)** Frequency at frontal ROI across 4 conditions in experiment 1 for **(i)** the "off" period, **(ii)** the "on" period, and **(iii)** the off period subtracted from the on period. **(B)** The same plots but for the parietal ROI in experiment 2. For all plots, mixed effects models were run: [frequency ~ condition + (1|Participant)]. Post hoc Wald tests were run for those which showed a statistically significant effect of condition ($p < 0.05$). * $p < 0.05$, ** $p < 0.01$, *** $p < 0.001$. Bars between conditions indicate differences between conditions, asterisks above conditions indicate a significant difference from zero, as per one-sample $t$ test. **Fig H. Autocorrelation of the inter stimulus interval in experiment 1.** Shown is the autocorrelation coefficient of inter-stimulus intervals across 20 lags, in which a value of 1 would indicate a perfectly periodic stimulus and lower values indicate a deviation from periodicity. The 4 different colours represent the 4 phase conditions and the error bars indicate standard error of the mean. It is clear that the stimulation we administered is far from periodic, with the autocorrelation coefficient quickly dropping to <0.1 after only 2 lags. We propose that if the sound was simply entraining itself, via the EEG, then we would see a highly periodic stimulus. This is not the case here. **Fig I. Stimulation-induced connectivity changes in alpha band (experiments 1 and 2).** **(Ai** and **Ei)** Topography of main effect of phase on average alpha band PLV for each channel as per ANOVA, for experiments 1 and 2, respectively. White marks indicate cluster-corrected $p < 0.05$. **(Aii** and **Eii)** Topography of variance between conditions of average alpha band PLI for each channel as per ANOVA, for experiments 1 and 2, respectively. White marks indicate cluster-corrected $p < 0.05$. **(Bi** and **Fi)** Stimulation-induced changes in alpha band PLV per condition as per $t$ test (compared to the "off" period), lines are plotted where $p < 0.01$. Red lines indicate an increase in the connectivity of that channel pair, blue lines indicate a decrease. **(Bii** and **Fii)** Stimulation-induced changes in alpha band PLI per condition as per $t$ test, lines are plotted where $p < 0.01$. Red lines indicate an increase in the connectivity of that channel pair, blue lines indicate a decrease. **(Ci** and **Gi)** Stimulation-induced changes in alpha band PLV per condition, collapsed across the stimulation period and across all other channels and the region of interest, in experiments 1 and 2, respectively. **(Cii** and **Gii)** Stimulation-induced changes in alpha band PLI per condition, collapsed across the stimulation period and across all other channels and the region of interest, in experiments 1 and 2, respectively. For all violin plots, mixed effects models were run: [connectivity ~ condition + (1|Participant)]. Post hoc Wald tests were run for those which showed a statistically significant effect of condition ($p < 0.05$). * $p < 0.05$, ** $p < 0.01$, *** $p < 0.001$. Bars between conditions indicate differences between conditions, asterisks above conditions indicate a significant difference from zero, as per one-sample $t$ test. **(Di** and **Hi)** Stimulation-induced changes in alpha band PLV per condition, collapsed across all other

channels and the region of interest across time, in experiments 1 and 2, respectively. (**Dii** and **Hii**) Stimulation-induced changes in alpha band PLI per condition, collapsed across all other channels and the region of interest across time, in experiments 1 and 2, respectively. **Fig J. Topography of permutation ANOVA stats for connectivity in each frequency band in experiment 1. (A)** Phase-locking value (PLV), **(B)** phase lag index (PLI). White dots show significant main effect of phase-targeted, cluster-corrected $p < 0.05$. **Fig K. Topography of permutation ANOVA stats for connectivity in each frequency band in experiment 2. (A)** Phase-locking value (PLV), **(B)** phase lag index (PLI). White dots show significant main effect of phase-targeted, cluster-corrected $p < 0.05$. **Fig L. Auditory evoked potential (AEP) and phase reset**. (**A** and **C**) Lowest pre-stimulus alpha power octile at Fz in experiment 3 (**A**) and Pz in experiment 4 (**C**). (**B** and **D**) Highest pre-stimulus alpha power octile at Fz in experiment 3 (**B**) and Pz in experiment 4 (**D**). (**A, B, C, D**) (**i**) Broadband (1–40 Hz) AEP*; (**ii**) Amplitude component of alpha band (7.5–12.5 Hz) endpoint-corrected Hilbert transformed AEP*; (**iii**) instantaneous phase of alpha band (7.5–12.5 Hz) from endpoint-corrected Hilbert transformed AEP**. Even at Pz, in experiment 4 there was post-stimuli phase alignment when the pre-stimulus alpha power was at its lowest amplitude (**Ciii**). * Black marks indicate ANOVA $p < 0.05$. ** Black marks indicate Rayleigh test $p < 0.05$, heatmap shows time series of Z-statistic. Fz and Pz electrodes from the hd-EEG system. **Fig M. Resultant per octile, and resultant vs. Z Stat for experiments 3 and 4**. (**Ai** and **Bi**) There was a clear linear relationship between alpha power octile and stimulus onset resultant in each condition, in both experiments. This meant that the phase was most consistent between trials in octile 8, and least consistent between trials in octile 1. We considered that this likely resulted from the use of 2 independent EEG systems, since the phase-locking (ecHT) system and the hd-EEG will be in greatest agreement, regarding phase, when alpha power is high, and stimulus onset is determined only by the ecHT. We suggest that the extent of the reset should not be dependent on this onset resultant. (**Aii** and **Bii**) The average phase angle was highly consistent across octiles for both experiments. (**C**) Here, we tested the extent by which the phase reset was related to the resultant at stimulus onset. We took 10,000 samples of 20 trials from each condition, computing stimulus onset resultant, averaging across conditions, and plotting this against auditory-evoked Z statistic. We found a statistically significant (in experiment 1, but not experiment 2), but very weak relationship ($R^2$ values <0.001), and hence confirmed our intuition that z-stat is not strongly dependent on onset resultant. **Table C. Number of trials per phase bin (experiments 3 and 4).** Phase estimates were used to sort all evoked potentials by stimulus onset phase, into ten 36˚ bins. *N* indicates number of participants. **Fig N. Phase accuracy plots for the ecHT electrode (in Fz) in the 3 conditions in Experiment 5.** Each line represents a participant, length of line indicates resultant (between 0 and 1). Phase accuracy is high in all 3 conditions. Phases are the same for pre-peak and sham. During sham, markers were recorded for each sound stimulus, but the volume was zero. **Fig O. Effects of stimulation phase on power and frequency at Fz for the whole duration of the nap opportunity in experiment 5.** Time-frequency representation of differences between stimulation conditions across the 30-min nap opportunity, as per paired *t* tests. Red colours indicate pre-peak>pre-trough and blue indicates pre-preak<pre-trough. Black lines outline statistically significant clusters. **Fig P. Inter-stimulus interval (ISI) and relation to individual alpha frequency for experiment 5. (A)** Sham-subtracted ISI vs. sham-subtracted alpha frequency. Lines and statistics derived from simple linear regression for each condition. The ISI between pulses of sounds showed a strong linear relationship with alpha frequency as observed in experiment 1 (**Fig 1D**). In a truly closed-loop experiment, it can become difficult to distinguish the direction of causality—did shorter ISI's speed up the brain's rhythms or did faster brain rhythms lead to shorter ISI's? Using the data from the sham condition, in which sound triggers were locked to pre-peak, but no sound was played, allowed

us to disentangle this conundrum. These dummy sound pulses were locked to the same phase as the pre-peak condition (**Fig N in S1 File**), but the ISIs differed, indicating that the ISI was indeed dependent on the brain's responses to sound, as opposed to the particular phase targeted. Furthermore, the distribution of ISIs differed between vigilance states further suggesting that the ISI and therefore the closed-loop algorithm used is dependent on the brain's physiology (see **Fig Q in S1 File**). **(B)** Average sham-subtracted ISI for pre-peak and pre-trough conditions. \*\*\* $p < 0.001$, \*\* $p < 0.01$, \* $p < 0.05$, $t$ tests. Blue represents pre-peak and green represents pre-trough. **Fig Q. Inter stimulus intervals per sleep stage.** ISIs of the sounds delivered in the pre-peak and pre-trough conditions and dummy sounds in the sham condition (i.e., no sound output) were investigated for each vigilance state (awake–W, N1 and N2 sleep). As participants transitioned from awake to N1 and N2 during experiment 5, the main frequencies detected in the phase-locking electrode also changed (alpha is more predominant during wakefulness, theta during N1 and sigma during N2). The shift in ISIs detected per vigilance stage for each stimulation condition indicates that the αCLAS algorithm followed brain activity. Violin plots show the percentage of ISI's in each frequency band, in each sleep stage, in each condition. Stats indicate output of linear mixed effects model [ISI_percentage ~ sleep_stage + (1|participant)]. \* $p < 0.05$, \*\* $p < 0.01$, \*\*\* $p < 0.001$. **Fig R. Cumulative N1 during experiment 5. (A)** Cumulative N1 sleep time series. Dashed line indicates end of stimulation. The plot shows that cumulative N1 sleep trailed off in pre-peak and sham conditions, as participants transitioned to deeper stages of sleep, while in pre-trough the amount of N1 continued to accumulate, with significantly more N1 sleep seen post-stimulation (see panel C). **(B)** Sham-subtracted time in N1 sleep collapsed across stimulation period. **(C)** Sham-subtracted time in N1 sleep collapsed across post-stimulation period. Note: statistics were run prior to sham subtraction. Post hoc comparisons were only carried out when a main effect of condition was seen in LME. Significance bars between violins indicate a difference between stimulation conditions, significance marks over violins indicate a difference from sham. \*\*\* $p < 0.001$, \*\* $p < 0.01$, \* $p < 0.05$, † $p < 0.1$, $t$ tests. Each violin shows a dot per participant, per condition, horizontal lines indicate the mean. **Fig S. eBOSC features per sleep stage for the sham condition.** Averages are shown for each feature, each region, each participant (coloured dots), for the sham condition. Stats indicate output of linear mixed effects model [eBOSC_feature ~ sleep_stage + (1|participant)]. \* $p < 0.05$, \*\* $p < 0.01$, \*\*\* $p < 0.001$. **Fig T. Time series for each eBOSC feature in experiment 5.** Average time series are shown for each feature, each region, for each condition. Data was smoothed using a 2-min moving mean window. Dashed line indicates end of stimulation. Time courses are represented in blue for pre-peak condition, green for pre-trough condition, and grey for sham condition. "Frontal" refers to the averaged data from channels F3 and F4, "Central" from channels C3 and C4, and "Occipital" from channels O1 and O2, all from the PSG system. **Fig U. All eBOSC features showing a main effect of condition. (A–I)** Time series of EEG eBOSC features (left) averaged across participants. Sham-subtracted features collapsed across stimulation period (middle) and post-stimulation period (right). "Frontal" refers to the averaged data from channels F3 and F4, "Central" from channels C3 and C4, and "Occipital" from channels O1 and O2, all from the PSG system. Note: Statistics were run prior to sham subtraction. Post hoc comparisons were only carried out when a main effect of condition was seen in LME. Significance bars between violins indicate a difference between stimulation conditions, significance marks over violins indicate a difference from sham. \*\*\* $p < 0.001$, \*\* $p < 0.01$, \* $p < 0.05$, † $p < 0.1$, $t$ tests. Each violin shows a dot per participant, per condition, horizontal lines indicate the mean. **Table D. Sleep scoring**. Duration in minutes of each vigilance state (wake, sleep N1, N2 and N3) for each participant, in each condition. Values are broken down into stimulation, post-stimulation, and whole nap periods. **Table E. eBOSC statistics from stimulation period. Table F. eBOSC statistics from**

**post-stimulation period.**
(DOCX)

## Acknowledgments

We thank Sarah Leslie and Adrian Cheung for their contributions to data collection, Giuseppe Atzori for scoring the sleep EEGs, David Wang for his technical support, and Ullrich Bartsch for fruitful discussion of our results.

## Author Contributions

**Conceptualization:** Henry Hebron, Nir Grossman, Derk-Jan Dijk, Ines R. Violante.

**Data curation:** Henry Hebron, Valeria Jaramillo, Lisa R. Yeh.

**Formal analysis:** Henry Hebron, Valeria Jaramillo, Edward Rhodes.

**Investigation:** Henry Hebron, Beatrice Lugli, Radost Dimitrova, Valeria Jaramillo, Lisa R. Yeh.

**Methodology:** Nir Grossman.

**Project administration:** Henry Hebron.

**Supervision:** Derk-Jan Dijk, Ines R. Violante.

**Validation:** Valeria Jaramillo.

**Visualization:** Henry Hebron.

**Writing – original draft:** Henry Hebron, Ines R. Violante.

**Writing – review & editing:** Henry Hebron, Beatrice Lugli, Radost Dimitrova, Valeria Jaramillo, Edward Rhodes, Nir Grossman, Derk-Jan Dijk, Ines R. Violante.

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
