## [Editor Report · Decision Letter 0]

31 Mar 2023

Dear Dr Hebron, 

Thank you for submitting your manuscript entitled "Alpha closed-loop auditory stimulation modulates waking alpha oscillations and sleep onset dynamics in a phase-dependent manner in humans" for consideration as a Research Article by PLOS Biology.

Your manuscript has now been evaluated by the PLOS Biology editorial staff as well as by an academic editor with relevant expertise and I am writing to let you know that we would like to send your submission out for external peer review.

Once your full submission is complete, your paper will undergo a series of checks in preparation for peer review. After your manuscript has passed the checks it will be sent out for review. To provide the metadata for your submission, please Login to Editorial Manager (https://www.editorialmanager.com/pbiology) within two working days, i.e. by Apr 04 2023 11:59PM.

Kind regards,

Luke

Lucas Smith, Ph.D.

Associate Editor

PLOS Biology

lsmith@plos.org

---

## [Decision Letter · Decision Letter 1]

27 Jun 2023

Dear Dr Hebron,

Thank you again for your patience while your manuscript "Alpha closed-loop auditory stimulation modulates waking alpha oscillations and sleep onset dynamics in a phase-dependent manner in humans" was peer-reviewed at PLOS Biology. It has now been evaluated by the PLOS Biology editors, an Academic Editor with relevant expertise, and by several independent reviewers. 

In light of the reviews, which you will find at the end of this email, we would like to invite you to revise the work to thoroughly address the reviewers' reports.

While the reviewers agree that the study is interesting, they have raised a number of important concerns that would need to be addressed before further consideration at PLOS Biology. The reviewers highlight that the writing is currently quite dense and inaccessible, and that the manuscript would need to be substantially revised to provide further context, justifications, explanations, and clarifications. We think that addressing these points is important to make the manuscript more accessible to the broad readership of PLOS Biology. In addition to thoroughly revising the writing of the study, it will be essential to thoroughly address the methodological issues raised by Reviewer 3, in order to add support to the conclusions of the study.

Given the extent of revision needed, we cannot make a decision about publication until we have seen the revised manuscript and your response to the reviewers' comments. Your revised manuscript is likely to be sent for further evaluation by all or a subset of the reviewers

**IMPORTANT - SUBMITTING YOUR REVISION**

*Re-submission Checklist*

*Published Peer Review*

*PLOS Data Policy*

*Blot and Gel Data Policy*

Sincerely,

Lucas

Lucas Smith, Ph.D.

Senior Editor

PLOS Biology

lsmith@plos.org

REVIEWS:

Reviewer #1: Summary: 

In a series of experiments using closed-loop auditory stimulation on awake adult humans, the authors explore the physiological effects of targeting alpha oscillations as measured at two different electrode sites (Pz, Fz, referenced to mastoid). The topic is interesting and timely, given recent interest in closed loop stimulation, and adds several pieces of information that will be helpful to others wishing to causally manipulate these higher frequency brain oscillations. While not definitive, the results are consistent with a phase-reset model of brain stimulation, and suggest some effects of manipulating alpha-band oscillations. 

The paper is extremely dense and must be deciphered, in part owing to its complex nature and in part owing to the structure and presentation of the work. There are many places in which the work could better convey the results and the authors' ideas more clearly and powerfully, and there are some places in which methods details and particularly justifications for design and analytic choices are needed (see suggestions). 

There is some theoretical backing for the work, though the framing is largely exploratory. There is room to better integrate the work with existing literature and ideas about how sound affects brain activity and the nature and roles of alpha oscillations. Given the journal's evaluation criteria, it might be better-suited to a specialized journal. 

Major: 

- the abstract and title ("alpha closed-loop auditory stimulation modulates (...) sleep onset dynamics in a phase-dependent manner in humans" ) as well as various statements of the findings ("we have shown the functional significance of this approach, during the transition to sleep") generates high expectation that the authors were able to successfully influence sleep onset. However, it seems that one of the stimulation conditions just prevented subjects from getting into deeper sleep as compared with the other stimulation condition, or sham. While this series of studies represents a valuable foray into causally manipulating alpha and trying to observe the physiological responses, the set up seems to detract from these efforts and makes the claim of results seem overstated.

- Density/readability suggestions:

a) There are about 5 "experiments" split over 3 "studies". The first encompasses experiments 1 and 2 (labelled as such), the second experiments 3 and 4 (also labelled), and the third study seems to have a fifth experiment (not called experiment 5?). The reader would have an easier time following this structure with a bit of an orientation as to the overall plan and goals and what each investigation looks at in the introduction, as well as consistent use of naming (like using 'experiment 5' for the last one). 

b) The rationale and meaning of some of the analyses is not always clearly explained when they are at first encountered, which in this methods-at-end format is already in the Results. For example, it was hard to follow why ISI is looked at in theta and sigma bands, and I did not understand why and on what basis the Pz and Fz were selected and followed through the various experiments. 

c) The article body has 8 large compound figures, many of which are made of smaller subplots - some as many as 47 (Figure 8) with a half-page of caption. Some of the plots are quite specialized and are likely to be unfamiliar graphic representations to many readers. I would suggest to select main questions of interest and move some of the other plots to Supplementary material, and/or find some way to more globally summarize the patterns in large numbers of similar subplots

d) Related, the figures could use more interpretable names / labels. There is quite some deciphering to be done for example to figure out how Figure 6. A, B, C, i - iv in Experiment 3 at Fz relate to similar plots for Exp 4 at Pz. 

- As the authors acknowledge, auditory evoked responses occur in response to sound and have a similar topography. There is considerable literature on the existence of an entrainment mechanism of auditory networks close to this frequency band (generally very slightly lower, like 4-9 Hz, though without hard boundaries). It was unclear to me to what degree the authors were able to separate an alpha modulation from a passing an AEP that might be composed of a normal evoked response and some addition from ongoing alpha oscillations through the band-pass filter, and how phase-resetting might be separated from a process of progressive entrainment. Is there some way of distinguishing these possibilities in these data?

- Why are the phase locking-values in Exp 2 and particularly Exp 1 so much lower using the hd-EEG system than with the ecHT device? (e.g., mean for Exp 1: 0.84 vs. 0.54) It is explained that the difference is because of the differences in EEG montages; however, evidence does not seem to be presented. Can the authors confirm by running the analysis on a re-referenced version of whichever set of electrodes is closest to Mastoid vs. Fz in their cap?

- Nap studies usually occur in the afternoon, for reasons relating to circadian rhythm and propensity to sleep. Why was Study 3 conducted in the morning, when subjects are most likely farthest from sleep? (Related, why were normal, healthy, well-rested subjects falling asleep in a few minutes with their eyes closed in the morning while loud clicks were played in Study 3?)

- More details and clarifications are needed as regards methods and justifications for design choices. For example, what was the justification for 80 dB (presumably SPL) sound presentation? This is considered quite loud for auditory experiments. How were the stimulation phases selected? (pre-peak 330 decrees, post-peak 60 degrees, etc.)

- More information is needed as regards sleep scoring - which system was used? Was the scorer blinded? (someone is acknowledged for scoring them and there is some discussion of finer-grained measures) 

- Why PLV and PLI? If PLI zero phase was discarded, does that mean PLV is potentially contaminated by volume conduction artifacts? Is that was is referred to when the authors describe relative sensitivity to type 1 and type 2 errors? Note that the meanings of these connectivity metrics is not referred to in the discussion; it is not clear what these analyses (and having included two measures) adds.

- Have the authors considered how changes in strength of the oscillations at different regions may have impacted the connectivity metrics (and their conclusions)? 

- How were sample sizes decided upon, which differ quite a bit between the experiments? (Justification/power analysis)

- Why did the authors decide to analyze connectivity in sensor space rather than source space (i.e., brain), which might have been more informative and is quite possible using hd-EEG?

- This paper format has results before methods. Keeping in mind that the reader does not yet have in mind the Methods details at time of reading the results, it would help to have a few more methods details included in the results, for example that there is to be post-peak and post-trough stimulation in Study 3, what electrode or electrodes are considered in the hd-EEG analysis, what the time window is for connectivity analysis following stimulation was, or what the analytic basis for statements like "phase-locking was highly accurate across experiments, greatest around locations targeted" are (in brief). What are the "locations targeted" here? Does this mean the scalp locations? Does it mean Fz and Pz separately? Why both?

- The explanation for using Fz vs Pz could use more explanation, and a stronger connection to extent literature on alpha origins and roles. The methods section says only "frontal and parietal regions of interest (ROI) were defined a priori" - but how/why/based on what? 

- In Exp 1, targeting Fz, an alpha-CLAS phase effect on power of alpha oscillations was found; this was not found for Pz, although a modulation of alpha power was found; the authors conclude that the effects of alpha-CLAS on alpha power and frequency are location-specific - what does this mean in terms of underlying anatomy? Why Fz and Pz to begin with?

- How far were the phases of Fz from those stimulated on Pz? What is the phase offset between these two scalp locations, and is it predictable or random? 

- Frequency differences - pg 10 - missing from this analysis is some indication of the degree to which a clear alpha peak is observed in each of these channels. This is important because theta-band activity is found just below the alpha band range, and is often not clearly distinguished from it. Sigma band activity related to spindles also only becomes clear and 'peaky' in N2. Theta is also stronger in frontal midline regions. There should be some discussion of how clearly the authors' analysis is getting at alpha as opposed to theta and sigma. 

- Is R2 = 0.97 in Fig 4 correct? Please explain in "Statistical Analyses" how this measure is computed here, as there are several methods (see https://jonlefcheck.net/2013/03/13/r2-for-linear-mixed-effects-models/), and (briefly) how it should be interpreted

- Why are different referencing strategies used across the studies and what are the potential effects on the reported results? 

- Fig 4A: the red to blue color scale is usually used to indicate a divergent scale where the two colors have opposite meanings. Here, blue is closer to 0, red is farther away, and white is at some arbitrary point in between (0.2). I suggest to use a continuous scale for easier interpretation.

- In some places, it is hard to follow the logic in the Results section; for example, the relative meanings of the "frequency difference" and "change in frequency difference" analyses presented in Figure 4 were not clear. 

- If alpha phase-dependent auditory stimulation influences the effect of sound on the brain through a process of inhibition (or put another way, relative cortical excitability), why did the authors not explicitly investigate the amplitude of cortical evoked auditory responses by alpha phase? Would the absence of an amplitude difference in the AEPs count against this theory? 

-

---

## [Editor Report · Decision Letter 2]

20 Feb 2024

Dear Dr Hebron,

Thank you for your patience while we considered your revised manuscript "Alpha closed-loop auditory stimulation modulates waking alpha oscillations and sleep onset dynamics in a phase-dependent manner in humans" for publication as a Research Article at PLOS Biology. This revised version of your manuscript has been evaluated by the PLOS Biology editors and the Academic Editor who is fully satisfied by the changed made in the revision.

Based on our Academic Editor's assessment of your revision, we are likely to accept this manuscript for publication. However, before we can editorially accept your study, we need you to address the following data and other policy-related requests.

**EDITORIAL REQUESTS: 

1) TITLE: We would like to suggest a minor change to the title, which we think will make it more broadly accessible and will highlight the approach developed here. If you agree, we suggest it be changed to something like: 

'A closed-loop auditory stimulation approach to selectively modulate alpha oscillations in the human brain.'

or maybe 

'A closed-loop auditory stimulation approach selectively modulates alpha oscillations and sleep onset dynamics in humans'

Happy for you to optimize further....

2) ETHICS STATEMENT: Thank you for including an ethics statement in your methods section. Please update this to include the approval number for the protocol approved by the University of Surrey Ethics Committee.

3) DATA and CODE: Thank you for providing the data and code related to your study on github. I took a look at this, and for some reason had a tough time finding the relevant data and code (I see some files related to Fig 1). Can you ensure that this has all relevant underlying data and code related to your manuscript? Please also update the Readme file to include a few more details about what the files are and how they relate to your manuscript. And as a last request, we ask that you generate a DOI for this dataset, to ensure its permanence. You can do that via zenodo (see: https://docs.github.com/en/repositories/archiving-a-github-repository/referencing-and-citing-content) 

We expect to receive your revised manuscript within two weeks. 

*Published Peer Review History*

*Press*

Sincerely,

Lucas

Lucas Smith, Ph.D.

Senior Editor

lsmith@plos.org

PLOS Biology

---

## [Editor Report · Decision Letter 3]

1 May 2024

Dear Dr Hebron,

Thank you for the submission of your revised Research Article "A closed-loop auditory stimulation approach selectively modulates alpha oscillations and sleep onset dynamics in humans" for publication in PLOS Biology and thank you for addressing our last editorial requests in this revision. On behalf of my colleagues and the Academic Editor, Simon Hanslmayr, I am pleased to say that we can in principle accept your manuscript for publication, provided you address any remaining formatting and reporting issues. These will be detailed in an email you should receive within 2-3 business days from our colleagues in the journal operations team; no action is required from you until then. Please note that we will not be able to formally accept your manuscript and schedule it for publication until you have completed any requested changes.

**Please note: as discussed over email, I have updated your 'data availability statement' to include the Zenodo DOI that you generated for the data and code provided on github. Please do take a quick look at this change to make sure everything is accurate. 

PRESS

Sincerely, 

Luke

Lucas Smith, Ph.D.

Senior Editor

PLOS Biology

lsmith@plos.org